# CoDiEmb: A Collaborative yet Distinct Framework for Unified Representation Learning in Information Retrieval and Semantic Textual Similarity

## Abstract

Obtaining text embeddings that excel across diverse downstream scenarios is a long-standing pursuit in representation learning, yet negative transfer remains a persistent obstacle. This challenge is particularly pronounced when jointly optimizing two core tasks: Information Retrieval (IR) and Semantic Textual Similarity (STS). Owing to discrepancies in data organization, text-length distributions, and evaluation metrics, naive co-training typically yields steep performance trade-offs. In this paper, we contend that systematically decoupling these tasks at both the design and training levels is essential for comprehensive model convergence. To this end, we propose CoDiEmb, a unified framework that processes IR and STS collaboratively yet distinctly. Unlike previous methods, CoDiEmb achieves superior performance under joint optimization without requiring complex multi-stage training pipelines or additional learnable components. CoDiEmb introduces three key innovations: (1) a unified data format compatible with inputs of any granularity. (2) task-specific objective functions aligned with evaluation metrics; and (3) a dynamic single-source data sampling strategy. Extensive experiments on 15 standard IR and STS benchmarks across three base encoders thoroughly validate the effectiveness of CoDiEmb. Our results and analysis demonstrate that the framework not only mitigates inter-task conflicts but also substantially alleviates the issues of anisotropy and over-smoothing in the semantic space. Our code is publicly available at `https://anonymous.4open.science/r/CoDiEmb`.

## 1 Introduction

Modern Natural Language Processing (NLP) is largely driven by two paradigms: generation and encoding (Muennighoff et al., 2024). The output of encoder models, known as text embeddings, represents a cornerstone of computational linguistics. Among the myriad applications and benchmarks for text embeddings, Semantic Textual Similarity (STS) and Information Retrieval (IR) stand out as two of the most critical (Gao et al., 2021). STS aims to determine the semantic proximity between two text segments, forming the foundation for technologies such as recommendation systems, text clustering, and content normalization (Sheng et al., 2024). IR, on the other hand, focuses on measuring the relevance between a query and a large document collection, playing a pivotal role in search engines, dialogue platforms, and AI agents (Sun et al., 2025).

Motivated by the goal of developing a universal text encoder proficient in both task families, state-of-the-art embedding models commonly train on large mixtures of STS and IR datasets using contrastive learning (Xiao et al., 2024; Lee et al., 2024a). While straightforward, this practice overlooks the inherent discrepancies between the two task types. Concretely, STS and IR exhibit significant differences in several key aspects:

- **Data Structure:** STS tasks typically organize data in triplets $(x_1, x_2, y)$, where the paired texts $x_1$ and $x_2$ are highly symmetric; swapping their positions does not alter the label $y$. Furthermore, because STS demands fine-grained semantic distinctions, $y$ often has multiple levels (e.g., 1 to 5). In contrast, IR datasets are inherently asymmetric, comprising a set

of queries $\{q\}_1^m$, a large document corpus $\{d\}_1^n$, and a relevance mapping $\{(q_i, d_j)\}_1^o$ that defines their relationships. During inference, a query $q_i$ is matched against each document in $\{d\}_1^n$, but only the pairs $(q_i, d_j)$ specified in the mapping are considered relevant. Moreover, as most IR tasks do not partition samples beyond positive and negative, their label granularity can be considered binary.

- **Text Length:** STS predominantly operates at the sentence level with short texts, perhaps because semantic similarity becomes ambiguous as length increases (Deshpande et al., 2023). Conversely, the queries and documents in IR tasks are highly flexible in length, with documents frequently spanning hundreds of tokens. As a result, although both tasks leverage cosine similarity for efficient matching, the underlying meaning of the calculation differs: STS prioritizes semantic equivalence, whereas IR leans towards topical or knowledge-level relevance.

- **Evaluation Metrics:** The primary metric for STS is Spearman's rank correlation coefficient (Zar, 2005), which measures the monotonic relationship between predicted and true rankings. The Normalized Discounted Cumulative Gain (nDCG) metric (Wang et al., 2013) used in IR is also list-wise but places greater emphasis on the correctness of top-ranked items. Furthermore, considering that documents relevant to a query are typically sparse in most IR tasks, nDCG@k is more commonly adopted.

These discrepancies lead to suboptimal performance when the two tasks are optimized indiscriminately. As we will demonstrate in Section 3, naively applying an objective function suited for one task, such as InfoNCE Loss (Oord et al., 2018) for IR or CoSENT Loss (Huang et al., 2024) for STS, is detrimental to the other. In contrast, our proposed framework, CoDiEmb, strikes a robust balance between IR and STS during collaborative training, approaching or even surpassing the performance of single-task fine-tuning.

Notably, some cutting-edge research has also observed these performance trade-offs. Asai et al. (2022) propose designing distinct instructions for different tasks and prepending them to the input text. While this strategy yields notable gains, the prior information provided by such instructions is limited and relies entirely on the model's implicit contextual understanding, lacking explicit gradient signals. Jina-embeddings-v3 (Sturua et al., 2024) introduces Task LoRA for parameter-level customization, but this necessitates storing a series of adapters. Moreover, if a document appears in $k$ task sets, it would require $k$ distinct embeddings, incurring prohibitive storage costs. NV Embed (Lee et al., 2024a) converts all data types into an IR format and constructs a two-stage training pipeline: first fine-tuning on IR datasets with hard negatives, followed by contrastive learning on a mixture of all corpora without them. This process inevitably discards a large volume of low-score STS data that cannot be formulated into positive pairs. In addition, as noted in prior work, coarse-grained contrastive objectives are ill-suited for tasks with fine-grained labels (Zhang & Li, 2024a;b).

This landscape reveals a pressing need for a unified, effective, and end-to-end solution for the joint optimization of IR and STS. To this end, we present CoDiEmb, a framework that **Co**llaboratively yet **Di**stinctly handles Information Retrieval and Semantic Textual Similarity across data formatting, loss functions, and sampling strategies.

Specifically, for IR tasks, we design a contrastive loss that supports multiple positives and hard negatives per anchor. This is augmented with cross-device negative sampling, which dramatically expands the pool of comparison candidates, yielding sharper separability. During this process, CoDiEmb's dynamic sampler ensures that, in each iteration, all GPUs draw samples strictly from disjoint subsets of the same data file, thereby providing pure task gradients. For STS tasks, rather than relying on the binary classification-style InfoNCE Loss or approximating the ranking objective by penalizing inverted pairs, we optimize directly for order consistency. Building on the Pearson Loss from Pcc-tuning (Zhang & Li, 2024a), we introduce our modified and adapted KL divergence Loss and PRO Loss (Peng et al., 2024), which substantially enhance the model's fine-grained semantic discrimination. Furthermore, to facilitate comprehensive convergence across all tasks, we allow dataset-specific batch sizes to fully balance their update counts.

In summary, the main contributions of this paper are as follows:

- We propose CoDiEmb, a framework that enables a model to converge effectively on both IR and STS tasks within a single training stage. CoDiEmb requires no adapter compo-

nents, and its unified data format is fully compatible with corpora of arbitrary granularity, eliminating the need to discard any samples.

- We formulate specialized loss functions tailored to the distinct characteristics of IR and STS. In conjunction with our custom dynamic sampler, this approach not only balances per-task iteration counts but also prevents the gradient interference induced by mixed-task batches.

- We conduct extensive experiments with MiniCPM (Hu et al., 2024), E5 (Wang et al., 2024), and BGE (Xiao et al., 2024) across 8 IR and 7 STS benchmarks, thoroughly validating the superiority of CoDiEmb. To further elucidate the underlying principles of our method, we provide a series of theoretical analyses, finding that CoDiEmb's joint optimization strategy effectively mitigates anisotropy (Ethayarajh, 2019) and over-smoothing (Shi et al., 2022) in the learned representation space.

## 2 METHODOLOGY

This section presents CoDiEmb (Figure 1), our end-to-end framework for unified representation learning across STS and IR. We begin in subsection 2.1 by introducing our task–agnostic data format, explaining its compatibility with inputs of heterogeneous granularity and its extensibility to other tasks. Building upon this, subsection 2.2 provides a detailed exposition of CoDiEmb's specialized loss functions, linking their design to the corresponding evaluation metrics. Finally, subsection 2.3 elaborates on our custom data sampler and the multi-device training configuration.

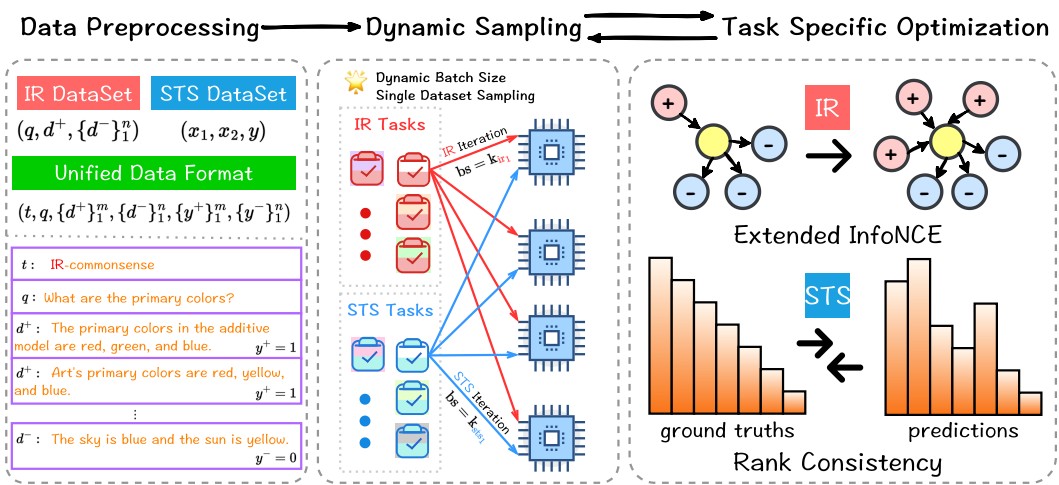

Figure 1: The overall workflow of CoDiEmb, covering data preprocessing, dynamic sampling, and multi-task joint optimization.

### 2.1 UNIFIED DATA FORMAT

As previously discussed, IR and STS tasks exhibit significant structural differences. IR samples are typically organized as tuples of $(q, d^+, \{d^-\})$, where $d^+$ is a document directly relevant to the query $q$, and $d^-$ is a set of hard negatives obtained through data mining. As this format does not include explicit scores, the supervisory signal is derived primarily from the annotator's binary partition of positive and negative samples. In contrast, for STS tasks, each text pair $(x_1, x_2)$ is associated with a numerical label $y$ that determines its relative ordering during inference. Furthermore, $y$ is not restricted to binary categories and can be an integer or floating-point number within any range. To accommodate both data types, CoDiEmb employs a unified format: $(t, q, \{d^+\}_1^m, \{d^-\}_1^n, \{y^+\}_1^m, \{y^-\}_1^n)$. Here, $t$ is a task identifier, which can be specified at the file level. In the subsequent training pipeline, samples with different identifiers are routed to distinct branches for task-specific processing. For fields absent in the original dataset, CoDiEmb fills them with default placeholders that are ignored during the forward pass, incurring no additional memory overhead.

This consolidated data structure is highly extensible. When handling STS tasks, we map the triplet $(x_1, x_2, y)$ to the query $q$, the first positive document $d_1^+$, and the first positive score $y_1^+$, respectively. For IR, we populate the query $q$, the positive set $\{d^+\}_1^m$, and the negative set $\{d^-\}_1^n$. Furthermore, data from classification or clustering tasks are also compatible with CoDiEmb. In these scenarios, the raw data can be partitioned by labels, allowing for intra-cluster (positive) and inter-cluster (negative) sampling to construct inputs for contrastive learning. An alternative strategy, adopted by works like Sentence-BERT (Reimers & Gurevych, 2019) and STS-Reg (Zhang & Li, 2024b), is the classifier-head architecture. In this case, the input text and its ground-truth label can be passed as $q$ and $y^+$, respectively.

Leveraging this unified data structure, CoDiEmb not only standardizes the loading of diverse corpora but also enables the configuration of differentiated loss functions tailored to task characteristics, thereby facilitating multi-granularity training. Although this paper focuses on the joint optimization of IR and STS, the potential of CoDiEmb extends beyond this scope. For instance, in Appendix A.3, we provide our implementation and test results on Pair Classification tasks.

## 2.2 Differentiated Loss Functions

As the optimization objective for model training, loss functions have profound impacts on neural network's performance. A well-designed loss should closely align with the evaluation metrics to provide effective learning signals. The primary metrics for IR and STS are nDCG@k and Spearman's correlation coefficient, respectively. Both are non-differentiable ranking metrics and thus cannot be directly optimized via backpropagation. For a given query $q$, let the top-$k$ documents retrieved by the model be $\{d_{\theta(1)}, d_{\theta(2)}, ..., d_{\theta(k)}\}$. The nDCG@k is calculated as follows:

$$\text{DCG@k} = \sum_{i=1}^{k} \frac{\text{rel}_i}{\log_2(i+1)} \quad \text{IDCG@k} = \sum_{i=1}^{k} \frac{\text{rel}_i^{\text{ideal}}}{\log_2(i+1)} \quad \text{nDCG@k} = \frac{\text{DCG@k}}{\text{IDCG@k}} \tag{1}$$

Here, $\text{rel}_i$ is the annotated relevance of $d_{\theta(i)}$, while $\text{rel}_i^{\text{ideal}}$ denotes the score of the ideal document at that rank; clearly, $\text{rel}_i \leq \text{rel}_i^{\text{ideal}}$. This formulation reveals that the core objective of nDCG@k is to place highly relevant documents at the top of the full candidate list. We analyze the average number of relevant documents per query across five open-source IR datasets, with results shown in Figure 2. It is evident that even within a vast corpus, content truly relevant to a specific query is typically sparse, making it feasible to enumerate most positive samples. Moreover, since mainstream IR datasets predominantly use binary labels, improving the nDCG@k for a query is equivalent to maximizing the predicted scores of its $m = \min(k, n\_\text{positives})$ positive documents. This objective aligns with the principles of contrastive learning but imposes two additional requirements: (1) documents involved in the relevance comparison should come from the same corpus and be as numerous as possible, and (2) a sufficient number of positive examples should be considered simultaneously.

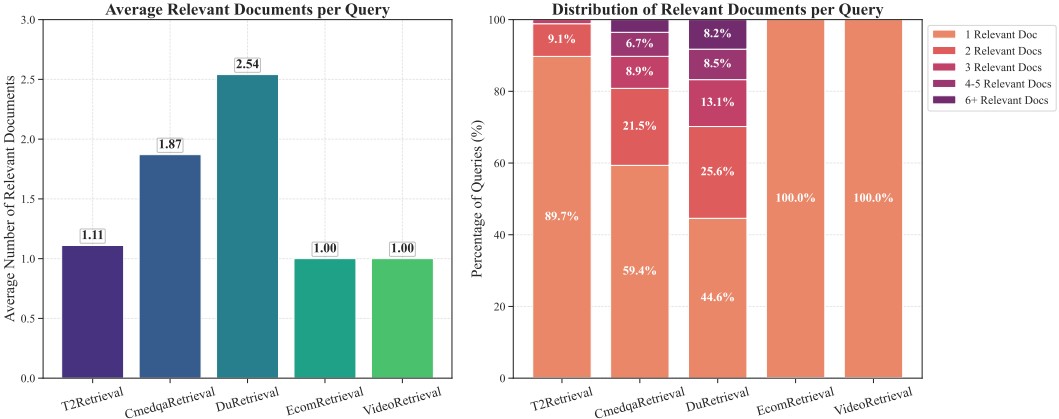

Figure 2: Average number and distribution of relevant documents per query across five widely used open-source IR datasets.

In CoDiEmb, the first requirement is primarily met by our custom sampler, which we will detail in subsection 2.3. The second is achieved through our design of an extended InfoNCE Loss that incorporates multiple positives and hard negatives. For a batch size of $N$, the loss is formulated as:

$$Z_i^+ = \sum_{j \neq i}^{N} \sum_{k=1}^{K^+} e^{\cos(q_i, d_{jk}^+)/\tau} \quad Z_i^- = \sum_{j=1}^{N} \sum_{k=1}^{K^-} e^{\cos(q_i, d_{jk}^-)/\tau}$$

$$\mathcal{L}_{\text{IR}} = -\mathbb{E}\left[\sum_{i=1}^{N} \sum_{c=1}^{K^+} \log \frac{e^{\cos(q_i, d_{ic}^+)/\tau}}{e^{\cos(q_i, d_{ic}^+)/\tau} + Z_i^+ + Z_i^-}\right] \quad (2)$$

In Equation 2, $K^+$ and $K^-$ denote the numbers of positive and hard negative examples, drawn from the input fields $\{d^+\}_1^m$ and $\{d^-\}_1^n$. If the available samples are fewer than $K^+$ or $K^-$, we sample with replacement. By considering multiple positives against an expanded set of negatives, this contrastive objective more closely approximates the nDCG@k metric, thereby boosting performance on IR tasks.

Unlike nDCG, which is a position-aware metric that assigns greater weight to top-ranked items, Spearman's correlation coefficient $\rho$ treats each sample equally and focuses on overall ranking quality. Its formula is described in Equation 3, where $n$ is the number of data points, and $d_i$ is the difference in ranks between the predicted and true scores for the $i$-th pair. Spearman's coefficient ranges from $-1$ to $1$, with higher values indicating stronger agreement between the model outputs and human judgments.

$$\rho = 1 - \frac{6 \sum_{i=1}^{n} d_i^2}{n(n^2 - 1)} \quad (3)$$

Training data for STS tasks often contain fine-grained annotation scores, for which coarse-grained modeling approaches like contrastive learning are suboptimal, as they fail to fully leverage such nuances and thus face a performance ceiling. To address this, Zhang & Li (2024a) introduced a Pearson Loss that directly optimizes the model at the rank level. CoDiEmb inherits this idea. Given a set of text pairs $\{(x_1^i, x_2^i)\}_{i=1}^n$, let the model's predicted cosine similarities be $X = \{\cos(f(x_1^i), f(x_2^i))\}_{i=1}^n$ and the list of ground-truth scores be $Y = \{y^i\}_{i=1}^n$. The Pearson Loss is calculated as:

$$r = \frac{\text{Cov}(X, Y)}{\sigma_X \sigma_Y} \quad \mathcal{L}_{\text{Pearson}} = -r + 1 \quad (4)$$

While effective, Pearson correlation primarily captures linear relationships. To model more complex mappings, CoDiEmb introduces two additional list-wise losses. The first adapts KL divergence, which measures the distance between a predicted distribution $Q$ and a target distribution $P$ as $\text{D}_{\text{KL}}(P||Q) = \sum_i p_i \log(p_i/q_i)$. An intuitive application to STS would involve converting predicted scores $\hat{y}$ and ground-truth scores $y$ into probability distributions ($q$ and $p$) using a standard Softmax function:

$$\hat{y}_i = \cos(f(x_1^i), f(x_2^i)) \quad q_i = \frac{\exp(\hat{y}_i/\tau)}{\sum_{j=1}^{N} \exp(\hat{y}_j/\tau)} \quad p_i = \frac{\exp(y_i/\tau)}{\sum_{j=1}^{N} \exp(y_j/\tau)} \quad (5)$$

Since $p_i$ is derived from ground-truth labels and carries no gradients, optimizing KL divergence is equivalent to minimizing cross-entropy. This process is analogous to knowledge distillation with soft labels and is logically sound. However, $p_i$ depends on the relative magnitudes of $y_i$ within the batch and can fluctuate substantially with the score distribution. Consider two batches: $Y_A = [0.9, 0.88, 0.2]$ and $Y_B = [0.6, 0.2, 0.1]$. With $\tau = 0.1$, we have $P_A = \text{Softmax}(Y_A) \approx [0.5496, 0.4499, 0.0005]$. Here, the first two samples account for 99.95% of the total probability mass, forcing the model to spend significant effort fitting the minute difference between 0.9 and 0.88, while the 0.2-scored sample receives a negligible gradient. Similarly, for batch $B$, $P_B = \text{Softmax}(Y_B) \approx [0.9756, 0.0179, 0.0066]$. In this case, the model is heavily incentivized to rank the first sample correctly, while the relative order of the other two is largely ignored.

This unstable weight allocation mechanism deviates from the spirit of Spearman correlation, which prioritizes rank consistency over absolute values. We therefore propose a Rank-normalized KL-divergence Loss $\mathcal{L}_{\text{RankKL}}$. Instead of comparing scores, we align the model's predictions with an

ideal distribution derived from ground-truth ranks. First, we sort all samples within a batch in descending order based on their labels to obtain ranks $r_i \in [0, N-1]$, where $N$ is the batch size. In case of ties, following the definition of Spearman correlation, $r_i$ is set to the average of their ranks. These ranks are then normalized to $y_i' \in [0, 1]$, matching their scale with predicted cosine similarities $\hat{y}$. We then define the target distribution $p_i$ as the Softmax of $y_i'$, while keeping $q_i$ as before. The final loss is:

$$p_i = \frac{\exp(y_i'/\tau)}{\sum_{j=1}^{N} \exp(y_j'/\tau)} \quad q_i = \frac{\exp(\hat{y}_i/\tau)}{\sum_{j=1}^{N} \exp(\hat{y}_j/\tau)}$$

$$y_i' = \frac{(N-1) - r_i}{N-1} \quad \mathcal{L}_{\text{RankKL}} = \sum_{i=1}^{N} p_i \log\left(\frac{p_i}{q_i}\right) \tag{6}$$

Compared to the original KL divergence, $\mathcal{L}_{\text{RankKL}}$ directly optimizes for rank, making it robust to the absolute magnitudes of ground-truth scores. This allows it to provide a stable gradient throughout training, driving the predicted ranking toward the desired order.

Building on this, we adapt Preference Rank Optimization (PRO), a reinforcement learning method originally from BEQUE (Peng et al., 2024) for query rewriting, to the domain of text representation. Similar to $\mathcal{L}_{\text{RankKL}}$, we first sort samples by their true scores $y_i$. For any pair $(i, j)$ in the sorted list where $i > j$, we define a weight $\mathcal{T}_i^j = \tau/(y_i - y_j)$, where $\tau$ is a temperature hyperparameter. We then set $\mathcal{T}_i^i$ to $\min_{i>j}(\mathcal{T}_i^j)$, i.e., the weight corresponding to the largest gap between sample $i$ and any subsequent sample. The $\mathcal{L}_{\text{PRO}}$ is formulated as below. It decomposes the ranking objective into $N-1$ sequential subproblems. For each anchor point $i$ in the list, we construct a classification task where the goal is to make its predicted score $\hat{y}_i$ higher than all subsequent items, with the optimization weights determined by their true similarity differences.

$$\mathcal{L}_{\text{PRO}} = -\mathbb{E}\left[\sum_{i=1}^{N-1} \log \frac{\exp(\hat{y}_i/\mathcal{T}_i^i)}{\sum_{j=i}^{N} \exp(\hat{y}_j/\mathcal{T}_i^j)}\right] \tag{7}$$

Finally, the total loss for STS tasks in CoDiEmb is a weighted sum of these components: $\mathcal{L}_{\text{STS}} = \alpha \mathcal{L}_{\text{Pearson}} + \beta \mathcal{L}_{\text{RankKL}} + \gamma \mathcal{L}_{\text{PRO}}$. During training, we alternate between $\mathcal{L}_{\text{IR}}$ and $\mathcal{L}_{\text{STS}}$ to update network parameters, preventing catastrophic forgetting and achieving a robust balance across tasks.

## 2.3 SAMPLER AND MULTI-GPU SETUP

As model and data volumes scale, distributed training has become standard practice in representation learning. Our previous analysis has highlighted that a core aspect of IR is making positive examples stand out from the entire document collection. Thus, with appropriate learning rates and iteration counts, a model's IR performance generally benefits from larger batch sizes (Zhang et al., 2022; Wu et al., 2022; Zhang et al., 2024a). Accordingly, CoDiEmb enables cross-device negative sampling when processing IR tasks to gather a larger pool of reference items.

However, merely increasing the sample count is insufficient for robust performance gains; the negatives obtained from other GPUs must be meaningful. In both real-world IR applications and benchmarks, a document is ranked against others from the same corpus. Therefore, negatives drawn from the same data distribution are more challenging and informative than random documents from a global pool. Consequently, CoDiEmb implements a custom data sampler that guarantees, within a single iteration, that each device processes non-overlapping shards of the same data file.

Conversely, for STS tasks, our empirical findings show that model convergence is not contingent on massive batch sizes. In fact, since many STS datasets use a small set of discrete integer labels (e.g., 0, 1, 2), an excessively large batch can lead to a high frequency of tied scores. Such a distribution can degrade the performance of rank-sensitive list-wise losses. Therefore, we disable cross-device sampling when processing STS batches.

Furthermore, the significant disparity in typical text lengths between IR (long documents) and STS (short sentences) makes a uniform batch size inefficient, leading to unbalanced GPU utilization and difficulty in managing per-task training iterations. To resolve this, CoDiEmb's data loader supports task-specific batch size configurations, optimizing training efficiency and providing finer control over the learning process.

## 3 EXPERIMENTS

### 3.1 MAIN RESULTS

Our experiments are primarily conducted on the well-established CMTEB leaderboard (Xiao et al., 2024), which comprises 7 STS tasks and 8 IR tasks spanning diverse domains such as news, medicine, finance and general knowledge. To validate the generality of our approach, we fine-tune three different PLMs as base encoders: MiniCPM-Embedding (Hu et al., 2024), multilingual-e5-large (Wang et al., 2024), and bge-large-zh-v1.5 (Xiao et al., 2024). For training, we adopt the publicly available CMTEB IR and STS datasets. Notably, three IR tasks—CovidRetrieval (Qiu et al., 2022), MMarcoRetrieval (Bonifacio et al., 2021), and MedicalRetrieval (Long et al., 2022)—do not provide dedicated training sets. Evaluations on these tasks are therefore performed in a zero-shot setting, which directly reflects the models' generalization capabilities. Detailed experimental configurations are provided in Appendix A.2.

Table 1 summarizes the overall performance of different methods on the full suite of CMTEB STS and IR tasks, while per-task scores are reported in Appendix A.4. To isolate the contributions of CoDiEmb's components, we compare it against several carefully designed baselines. Here, "InfoNCE" denotes training solely with the InfoNCE Loss. For STS tasks under this setting, pairs with low similarity are filtered via a threshold to ensure the correctness of contrastive objectives. Conversely, "CoSENT" refers to training exclusively with the CoSENT Loss, a prevalent approach in STS research (Li et al., 2024; Yu et al., 2025). The formulas for both objectives are given below, where the notation is consistent with subsection 2.2. Additionally, "Mixed" indicates the adoption of a mixed-batch sampler during training. While this sampler still requires that texts within each GPU originate from the same data file, it places no such restriction across GPUs. Consequently, in a single iteration, different GPUs may process different task types, providing the model with mixed-task gradients.

$$\mathcal{L}_{\text{InfoNCE}} = -\mathbb{E}\left[\sum_{i=1}^{N} \log \frac{1_{\text{label}} e^{\cos\left(f(x_1^i), f(x_2^i)\right)/\tau}}{\sum_{j=1}^{N} e^{\cos\left(f(x_1^i), f(x_2^j)\right)/\tau}}\right] \tag{8}$$

$$\mathcal{L}_{\text{Cosent}} = \log\left(1 + \sum 1_{y_i > y_j} \exp\left(\cos\left(f(x_1^j), f(x_2^j)\right)/\tau - \cos\left(f(x_1^i), f(x_2^i)\right)/\tau\right)\right)$$

Table 1: Main results on the CMTEB benchmark. Avg. IR and Avg. STS report the average nDCG@10 across 8 IR tasks and the average Spearman correlation across 7 STS tasks, respectively. The Overall Score equals the sum of these two metrics.

| Methods | PLMs | Avg. IR | Avg. STS | Overall Score |
|---------|------|---------|----------|---------------|
| InfoNCE | MiniCPM | 74.23 | 60.53 | 134.76 |
| CoSENT | MiniCPM | 71.30 | 70.05 | 141.35 |
| Mixed | MiniCPM | 73.05 | 70.32 | 143.37 |
| CoDiEmb | MiniCPM | **75.73** | **71.15** | **146.88** |
| InfoNCE | e5-large | **70.90** | 56.32 | 127.22 |
| CoSENT | e5-large | 65.69 | 64.61 | 130.30 |
| Mixed | e5-large | 68.61 | 66.89 | 135.50 |
| CoDiEmb | e5-large | 70.62 | **68.19** | **138.81** |
| InfoNCE | bge-large | **71.73** | 58.09 | 129.82 |
| CoSENT | bge-large | 66.55 | 64.70 | 131.25 |
| Mixed | bge-large | 68.67 | 67.12 | 135.79 |
| CoDiEmb | bge-large | 71.07 | **67.87** | **138.94** |

As demonstrated in Table 1, compared to using InfoNCE alone, CoDiEmb attains comparable or superior IR performance while consistently outperforming it on STS tasks, leading to a significantly higher overall score across all backbones. This phenomenon is interpretable: when harnessing a unified contrastive learning approach, the threshold-filtered STS samples steer the model toward cluster-oriented updates akin to IR, thus effectively acting as a form of data augmentation for the IR task. However, this slight improvement in IR comes at the expense of a drastic degradation in STS

performance. For this reason, CoDiEmb avoids relying on coarse-grained contrastive learning as the primary optimization strategy for STS.

When benchmarked against the CoSENT-only approach, CoDiEmb demonstrates markedly superior performance on both IR and STS tasks. For instance, with multilingual-e5-large as the backbone, CoDiEmb achieves gains of 4.93 on average nDCG@10 and 3.58 on average Spearman correlation—substantial increases for both metrics. This highlights the inadequacy of a single pair-wise ranking loss for the distinct optimization of both tasks. A similar pattern holds when comparing against the mixed-gradient sampler. By strictly ensuring that each GPU processes a disjoint subset of the same dataset per iteration while flexibly balancing the update frequencies of different data sources, CoDiEmb delivers steady improvements across tasks. Collectively, these observations confirm the effectiveness of CoDiEmb's specialized loss design and its single-source sampling strategy. Furthermore, we also provide robustness experiments under various batch size configurations as well as ablation studies of the loss functions in Appendix A.5 and Appendix A.6.

### 3.2 COMPARISON WITH SINGLE-TASK MODELS

To conclusively substantiate that CoDiEmb achieves a synergistic balance rather than a simple trade-off, we compare it against two specialist models: one trained exclusively on IR data (IR-only) and another on STS data (STS-only). The results, presented in Table 2, first reveal the severe limitations of single-task training. While the specialist models excel on their native tasks, they suffer a catastrophic performance collapse when transferred to the other domain. On average, the IR-only model's STS performance is 17.3 points lower than the STS specialist, while the STS-only model's IR performance is a staggering 22.2 points lower than its IR counterpart.

In stark contrast, CoDiEmb demonstrates a highly effective and favorable trade-off. It incurs only a marginal cost on IR tasks (averaging -0.71 points relative to the IR specialist), while delivering substantial gains on STS tasks (improving by 1.92 points on average over the STS specialist). More strikingly, CoDiEmb consistently outperforms the STS-only model on its own primary task across all backbones, indicating that the STS task does not suffer from negative transfer but instead benefits from co-training with IR data under our collaborative-yet-distinct paradigm.

Table 2: Performance comparison between CoDiEmb and single-task specialist models. Values in parentheses denote the performance difference ($\Delta$) of CoDiEmb relative to the corresponding specialist model, indicating a performance gain (+) or cost (-).

| PLMs | Method | Avg. IR | Avg. STS | Overall Score |
|---|---|---|---|---|
| MiniCPM-Embedding | IR-only | 76.10 | 49.67 | 125.77 |
| | STS-only | 62.28 | 68.83 | 131.11 |
| | CoDiEmb | 75.73 (-0.37) | 71.15 (+2.32) | **146.88** |
| multilingual-e5-large | IR-only | 71.34 | 48.22 | 119.56 |
| | STS-only | 48.02 | 66.37 | 114.39 |
| | CoDiEmb | 70.62 (-0.72) | 68.19 (+1.82) | **138.81** |
| bge-large-zh-v1.5 | IR-only | 72.12 | 51.72 | 123.84 |
| | STS-only | 45.64 | 66.26 | 111.90 |
| | CoDiEmb | 71.07 (-1.05) | 67.87 (+1.61) | **138.94** |

## 4 ANALYSIS

To assess the intrinsic quality of the embedding space learned by CoDiEmb, we move beyond benchmark scores to conduct an in-depth quantitative analysis of its geometric properties. A high-quality embedding space should capture subtle semantic distinctions while maintaining a well-dispersed distribution, thereby maximizing representational capacity. However, prior work has pointed out that pre-trained language models commonly suffer from over-smoothing (Shi et al., 2022) and anisotropy (Ethayarajh, 2019), both of which can severely compromise representation quality.

Over-smoothing arises when a model loses the ability to differentiate among tokens within a text, mapping them to overly similar embeddings. Anisotropy, by contrast, occurs when embeddings

collapse into a narrow cone in the vector space, resulting in limited expressiveness. Our central hypothesis is that CoDiEmb—through its collaborative architecture, task-specific loss functions, and task-pure gradient signals—effectively mitigates these issues. To validate this hypothesis, we employ a suite of diagnostic metrics established in prior work (Zhang et al., 2025a) to evaluate the health of the learned embedding space.

Given an input sequence $T = [t_1, t_2, \ldots, t_n]$, the model outputs a token embedding matrix $X \in \mathbb{R}^{n \times d}$ whose rows are the token vectors $\{x_1, x_2, \ldots, x_n\}$. We quantify over-smoothing using Token-wise Similarity (TokSim), defined as the average pairwise cosine similarity among distinct tokens: $\text{TokSim}(X) = \frac{1}{n(n-1)} \sum_{i \neq j} \frac{x_i^T x_j}{\|x_i\|_2 \|x_j\|_2}$. A lower value indicates better separability.

To evaluate anisotropy, we analyze the singular value spectrum of $X$. First, we report the matrix rank $\text{rank}(X)$, where a higher value indicates richer, less redundant information. We then perform Singular Value Decomposition (SVD) on $X$ and adopt two additional indicators. The Condition Number, $\kappa(X) = \sigma_{\max}/\sigma_{\min}$, is the ratio of the largest to the smallest singular value. A lower value is preferred, as it signifies a more uniform spectrum. The SVD Entropy $H(X)$ measures the richness of the effective semantic dimensions. It is calculated by first normalizing the squared singular values into a probability distribution $p_i = \sigma_i^2 / \sum_{j=1}^{k} \sigma_j^2$ and then computing the entropy $H(X) = -\sum_{i=1}^{k} p_i \log(p_i)$. A higher value indicates that more semantic dimensions contribute meaningfully to the representation, signaling a lower degree of anisotropy.

We compute these four metrics on the seven STS test sets in CMTEB using the BGE backbone. As shown in Table 3, CoDiEmb exhibits a consistent and significant advantage across all metrics. It achieves the lowest token-wise similarity, confirming its effectiveness in mitigating over-smoothing. Concurrently, it systematically obtains a higher rank and SVD entropy, alongside a markedly lower condition number. These results provide strong quantitative evidence that CoDiEmb produces a more expressive, isotropic, and well-structured embedding space.

Table 3: Analysis of embedding space properties on the CMTEB STS test sets. For Rank and SVD Entropy, higher is better. For Token Similarity and Condition Number, lower is better.

| Method | Metric | ATEC | BQ | LCQMC | PAWSX | STSB | AFQMC | QBQTC | Avg |
|---|---|---|---|---|---|---|---|---|---|
| InfoNCE | Rank | 14.52 | 12.58 | 10.61 | 38.41 | 19.54 | 14.36 | 9.35 | 17.05 |
| | Token Similarity | 73.22 | 78.79 | 70.69 | 70.67 | 77.78 | 72.41 | 72.92 | 73.93 |
| | SVD Entropy | 1.61 | 1.27 | **1.62** | 2.10 | 1.49 | 1.60 | 1.45 | 1.59 |
| | Condition Number | 21028.14 | 21571.02 | 22738.23 | 1245.10 | 7025.74 | 24150.67 | 20942.72 | 16957.37 |
| CoSENT | Rank | 14.57 | 12.68 | 10.62 | 38.44 | 19.67 | 14.40 | 9.38 | 17.11 |
| | Token Similarity | 73.04 | 76.30 | 74.55 | 69.97 | 75.32 | 73.36 | 74.02 | 73.79 |
| | SVD Entropy | 1.58 | 1.36 | 1.42 | 2.07 | 1.58 | 1.56 | 1.38 | 1.56 |
| | Condition Number | 17917.04 | 16278.22 | 22119.27 | 348.73 | 3742.57 | 20770.19 | 19353.22 | 14361.32 |
| Mixed | Rank | 14.45 | 12.58 | 10.61 | 38.42 | 19.52 | 14.33 | 9.35 | 17.04 |
| | Token Similarity | 67.83 | 74.91 | 71.86 | **61.58** | 66.54 | 67.93 | 70.46 | 68.73 |
| | SVD Entropy | 1.77 | 1.40 | 1.54 | 2.47 | 1.96 | 1.76 | 1.53 | 1.78 |
| | Condition Number | 19580.63 | 17888.89 | 20916.23 | 1007.10 | 6118.76 | 21357.64 | 18648.04 | 15073.90 |
| CoDiEmb | Rank | **14.97** | **12.85** | **10.63** | **38.44** | **19.80** | **14.62** | **9.47** | **17.25** |
| | Token Similarity | **67.67** | **72.92** | **70.61** | 61.68 | **65.77** | **67.82** | **70.22** | **68.10** |
| | SVD Entropy | **1.81** | **1.50** | 1.59 | **2.49** | **2.01** | **1.79** | **1.54** | **1.82** |
| | Condition Number | **7413.04** | **10847.82** | **19901.18** | **265.59** | **507.70** | **13129.53** | **15584.98** | **9664.26** |

## 5 CONCLUSION

In this paper, we introduced CoDiEmb, a unified training framework that optimizes text embeddings for Information Retrieval (IR) and Semantic Textual Similarity (STS) in a collaborative-yet-distinct manner. Through innovations in data formatting, loss design, and sampling strategies, CoDiEmb delivers significant performance gains across a broad range of tasks. The success of CoDiEmb suggests that the pursuit of universal text encoders should transcend conventional multi-stage contrastive learning. Instead, a more promising direction lies in developing a unified framework that explicitly leverages task-specific characteristics to attain a synergistic equilibrium. Future work will focus on extending CoDiEmb and exploring its generalization to a broader range of tasks.

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

# A APPENDIX

## A.1 RELATED WORK

As a cornerstone of computational linguistics, text representation has long attracted sustained attention from the research community. Broadly, the evolution of this field can be divided into three distinct stages. Early works, such as Word2Vec (Mikolov et al., 2013) and GloVe (Pennington et al., 2014), focused on lexical-level representations, primarily yielding context-independent encodings. The advent of Pre-trained Language Models (PLMs), exemplified by BERT (Devlin et al., 2019), marked a paradigm shift in NLP, driven by significant advances in computational power and model architectures (Vaswani et al., 2017). Pioneering studies including Sentence-BERT (Reimers & Gurevych, 2019), BERT-flow (Li et al., 2020), and SimCSE (Gao et al., 2021) subsequently integrated discriminative PLMs with representation learning from different perspectives, achieving notable breakthroughs in sentence-level embeddings. This period was characterized by rapid progress in unsupervised text representation. Leveraging innovative data augmentation techniques, approaches like SNCSE (Wang & Dou, 2023), PromptBERT (Jiang et al., 2022), and CoT-BERT (Zhang et al., 2024a) attained strong performance on the seven STS tasks aggregated by SentEval (Conneau & Kiela, 2018), using only unsupervised corpora composed of individual sentences.

Despite the powerful semantic understanding afforded by bidirectional attention, the architectural constraints of BERT-style models—namely, a 512-token maximum sequence length and relatively modest parameter counts—long prevented text embedding research from fully benefiting from the scaling laws of Large Language Models (LLMs). To address this, PromptEOL (Jiang et al., 2024) pioneered the application of 7B-scale generative models for text representation and proposed the widely adopted Explicit One-Word Limitation (EOL) hypothesis. Building on this, Zhang et al. (2024b) demonstrated through extensive experiments that EOL is unnecessary for fine-tuning and introduced two simple yet effective techniques that directly enhance the representational capacity of base models. Subsequently, Pcc-tuning (Zhang & Li, 2024a) provided an insightful analysis of the binary classification nature of contrastive learning, explaining the performance bottlenecks it faces in fine-grained semantic discrimination, and proposed directly optimizing the Pearson correlation coefficient.

With the rise of Retrieval-Augmented Generation (RAG) and AI Agent systems, a consensus has formed around the need for powerful, general-purpose text encoders. In response, models such as E5 (Wang et al., 2022), GTE (Li et al., 2023), BGE (Xiao et al., 2024), and Nomic Embed (Nussbaum et al., 2024) have been trained on massive datasets, typically employing multi-stage

contrastive learning pipelines to progressively refine semantic expressiveness. State-of-the-art embedding models, including Gecko (Lee et al., 2024b), Gemini Embedding (Lee et al., 2025), and Qwen Embedding (Zhang et al., 2025b), can be viewed as continuations of this paradigm.

CoDiEmb builds upon these prior works, aiming to learn multi-functional representations within a single, unified framework. However, our approach diverges by advocating for a deeper, task-aware analysis and customization of the training process. Our results suggest that tailoring optimization to each task's distinctive properties can enhance, rather than compromise, the model's overall versatility. We hope this work will inspire further research into developing specialized, yet synergistic, optimization strategies for universal representation learning.

## A.2 IMPLEMENTATION DETAILS

This appendix provides supplementary implementation details for the experiments in Section 3. As noted in the main text, both our training and evaluation data are drawn from the CMTEB benchmark. Detailed statistics for each dataset are provided in Table 4. In particular, for the T2Retrieval task (Xie et al., 2023), we mined a set of hard negatives for each query to ensure parity with corpora that naturally include them. Furthermore, following established practices, we designed task-specific instructions and prepended them to each input text before encoding. The datasets and instructions used are identical across all methods and experiments to ensure a fair comparison.

For MiniCPM-Embedding and multilingual-e5-large, we derive text representations via mean pooling and mask out instruction tokens. For bge-large-zh-v1.5, we follow the official guidelines and employ CLS pooling. To conserve computational resources, all training runs leverage DeepSpeed ZeRO-1 and gradient checkpointing.

Table 4: Overview and statistics of the CMTEB IR and STS datasets. A dash (-) in the "Train" column indicates that the task lacks a training split; evaluations on these tasks are therefore conducted in a zero-shot setting.

| Name | Type | #Train | #Test | Description |
|---|---|---|---|---|
| CmedqaRetrieval (Qiu et al., 2022) | Retrieval | 99,904 | 4,000 | Online medical consultation texts |
| CovidRetrieval (Qiu et al., 2022) | Retrieval | - | 949 | The COVID-19 news article retrieval dataset |
| DuRetrieval (Qiu et al., 2022) | Retrieval | 83,456 | 2,000 | A large-scale Chinese web search engine paragraph retrieval benchmark |
| MMarcoRetrieval (Bonifacio et al., 2021) | Retrieval | - | 6,980 | the multilingual version of the MS MARCO paragraph ranking dataset |
| T2Retrieval (Xie et al., 2023) | Retrieval | 698,752 | 22,800 | T2Ranking: A large-scale Chinese paragraph ranking benchmark |
| EcomRetrieval (Long et al., 2022) | Retrieval | 81,920 | 1,000 | Multi-CPR: A Multi Domain Chinese Dataset for Passage Retrieval |
| MedicalRetrieval (Long et al., 2022) | Retrieval | - | 1,000 | Multi-CPR: A Multi Domain Chinese Dataset for Passage Retrieval |
| VideoRetrieval (Long et al., 2022) | Retrieval | 82,560 | 1,000 | Multi-CPR: A Multi Domain Chinese Dataset for Passage Retrieval |
| AFQMC (Xu et al., 2020) | STS | 34,334 | 3,861 | Ant Financial Question Matching Corpus |
| ATEC (Xiao et al., 2024) | STS | 62,477 | 20,000 | ATEC NLP Sentence Pair Similarity Competition |
| BQ (Chen et al., 2018) | STS | 100,000 | 10,000 | Banking Question Semantic Similarity |
| LCQMC (Liu et al., 2018) | STS | 238,766 | 12,500 | Large-scale Chinese Question Matching Corpus |
| PAWSX (Yang et al., 2019) | STS | 49,129 | 2,000 | Translated PAWS evaluation pairs |
| QBQTC (Xu et al., 2020) | STS | 180,000 | 5,000 | QQ Browser Query Title Corpus |
| STSB (Cer et al., 2017) | STS | 5,231 | 1,360 | Translated STS-B into Chinese |

## A.3 EXTENDING CODIEMB TO PAIR CLASSIFICATION

In this appendix, we investigate the transferability of our CoDiEmb framework by extending it to Pair Classification (PC) tasks. The CMTEB benchmark includes two PC datasets, OCNLI (Hu et al., 2020) and CMNLI (Xu et al., 2020), whose data formats are largely consistent with those of STS. For methodological simplicity, we handle PC data via the same processing pipeline and objective functions designed for STS. We conduct two experiments using the bge-large-zh-v1.5 backbone: first, we train a model on IR and STS data only and evaluate its zero-shot transfer performance on PC; second, we train a model on all three task types. The results are presented in Table 5.

As shown in Table 5, joint training on IR and STS corpora enables the model to not only significantly outperform the raw baseline on those primary tasks but also to achieve a discernible performance gain on the unseen PC task (65.18 vs. 62.82). This suggests that CoDiEmb's training paradigm improves the fundamental geometric properties of the embedding space, leading to enhanced generalization. This aligns with our analysis in Section 4, where we demonstrated that CoDiEmb mitigates the issues of over-smoothing and anisotropy.

Furthermore, by explicitly incorporating PC data into the training regimen, CoDiEmb achieves a substantial boost in PC performance—an increase of over 23 accuracy points—with only a minimal trade-off in its IR and STS capabilities. Collectively, these findings demonstrate that the CoDiEmb framework can be effectively extended to new task domains, underscoring its versatility and robustness.

Table 5: Performance of CoDiEmb when extended to include Pair Classification (PC) tasks. All experiments adopt the bge-large-zh-v1.5 backbone. Avg. PC reports the average accuracy on the OCNLI and CMNLI datasets.

| PLMs | Method | Avg. IR | Avg. STS | Avg. PC | Overall Score |
|---|---|---|---|---|---|
| | Raw Model | 47.74 | 50.57 | 62.82 | 161.13 |
| bge-large-zh-v1.5 | CoDiEmb (IR + STS) | 71.07 | 67.87 | 65.18 | 204.12 |
| | CoDiEmb (IR + STS + PC) | 70.89 | 67.13 | 88.62 | **226.64** |

## A.4 PER-TASK RESULTS FOR STS AND IR

In subsection 3.1, we presented the aggregate performance of CoDiEmb and the baselines on the IR and STS task families. Here, we report detailed per-task scores for each method on the 8 IR and 7 STS tasks from the CMTEB benchmark. The results are shown in Table 7 and Table 6, respectively. Consistent with standard practice in the field, we report nDCG@10 as the primary metric for IR and Spearman's rank correlation for STS.

Table 6: Spearman's correlation scores of different methods on the 7 STS tasks in CMTEB. The last two columns, Avg. IR and Avg. STS, represent the model's average performance on IR and STS, respectively. Corresponding IR results are available in Table 7.

| Methods | AFQMC | ATEC | BQ | LCQMC | PAWSX | QBQTC | STS-B | Avg. IR | Avg. STS |
|---|---|---|---|---|---|---|---|---|---|
| *Implementation on MiniCPM-Embedding* | | | | | | | | | |
| InfoNCE | 61.51 | 58.03 | 67.78 | 71.89 | 40.93 | 41.82 | 81.73 | 74.23 | 60.53 |
| CoSENT | 69.28 | 59.54 | 73.57 | 79.97 | 63.95 | 58.35 | 85.69 | 71.30 | 70.05 |
| Mixed | 70.77 | 61.37 | 72.01 | 78.40 | 65.48 | 59.29 | 84.93 | 73.05 | 70.32 |
| CoDiEmb | 69.70 | 60.56 | 74.23 | 80.38 | 67.12 | 60.98 | 85.11 | **75.73** | **71.15** |
| *Implementation on multilingual-e5-large* | | | | | | | | | |
| InfoNCE | 52.07 | 53.12 | 69.72 | 72.83 | 26.99 | 40.46 | 79.02 | **70.90** | 56.32 |
| CoSENT | 53.18 | 53.09 | 72.19 | 80.29 | 57.53 | 53.52 | 82.50 | 65.69 | 64.61 |
| Mixed | 58.36 | 54.91 | 72.83 | 79.99 | 63.44 | 56.87 | 81.84 | 68.61 | 66.89 |
| CoDiEmb | 62.70 | 55.65 | 73.16 | 80.44 | 66.17 | 56.43 | 82.78 | 70.62 | **68.19** |
| *Implementation on bge-large-zh-v1.5* | | | | | | | | | |
| InfoNCE | 54.47 | 54.34 | 68.64 | 74.16 | 34.31 | 41.12 | 79.61 | **71.73** | 58.09 |
| CoSENT | 56.73 | 54.52 | 72.55 | 80.54 | 55.34 | 52.56 | 80.67 | 66.55 | 64.70 |
| Mixed | 62.59 | 56.59 | 73.04 | 80.16 | 59.51 | 56.82 | 81.13 | 68.67 | 67.12 |
| CoDiEmb | 66.60 | 58.32 | 72.27 | 80.07 | 60.48 | 56.90 | 80.45 | 71.07 | **67.87** |

## A.5 THE IMPACT OF BATCH SIZE ON MODEL PERFORMANCE

Performance sensitivity to hyperparameters is a persistent challenge in deep learning. This issue is particularly acute in representation learning, which often involves large-scale distributed training. In the context of this paper, each query from an IR dataset is paired with $K^+$ positives and $K^-$ hard negatives. Consequently, a change of $m$ in the number of queries per device batch results in a substantial fluctuation of $\text{n\_gpu} \times m \times (K^+ + K^-)$ in the total number of contrastive examples per optimization step, where n_gpu is the number of GPUs (32 in our case). This dynamic implies that even a modest adjustment to the batch size dramatically alters the training landscape. While a larger set of in-batch negatives can enhance the discriminative power of the learned embeddings, it also increases the risk of introducing false negatives, making the final performance difficult to predict.

A desirable property of a general-purpose framework like CoDiEmb is robustness to such variations, ensuring stable and predictable outcomes. To evaluate this, we test the performance of CoDiEmb

Table 7: nDCG@10 scores of different methods on the 8 IR tasks in CMTEB. The last two columns, Avg. IR and Avg. STS, represent the model's average performance on IR and STS, respectively. Corresponding STS results are available in Table 6.

| Methods | Cmedqa | Covid | Du | Ecom | MMarco | Medical | T2 | Video | Avg. IR | Avg. STS |
|---------|--------|-------|-----|------|--------|---------|-----|-------|---------|----------|
| *Implementation on MiniCPM-Embedding* | | | | | | | | | | |
| InfoNCE | 41.99 | 90.73 | 88.78 | 65.42 | 83.76 | 61.26 | 86.91 | 74.98 | 74.23 | 60.53 |
| CoSENT | 42.28 | 81.81 | 86.70 | 65.66 | 78.89 | 59.57 | 84.52 | 70.97 | 71.30 | 70.05 |
| Mixed | 41.82 | 90.01 | 87.62 | 64.10 | 83.21 | 59.64 | 86.22 | 71.75 | 73.05 | 70.32 |
| CoDiEmb | 45.43 | 90.61 | 89.51 | 69.24 | 84.26 | 62.86 | 87.36 | 76.55 | **75.73** | **71.15** |
| *Implementation on multilingual-e5-large* | | | | | | | | | | |
| InfoNCE | 41.85 | 74.97 | 85.90 | 65.76 | 77.40 | 59.82 | 84.47 | 77.00 | **70.90** | 56.32 |
| CoSENT | 33.49 | 70.31 | 83.82 | 62.50 | 73.05 | 51.66 | 82.54 | 68.17 | 65.69 | 64.61 |
| Mixed | 38.53 | 75.79 | 83.02 | 63.64 | 75.75 | 55.92 | 82.12 | 74.07 | 68.61 | 66.89 |
| CoDiEmb | 41.92 | 76.20 | 85.85 | 66.31 | 76.95 | 56.80 | 84.30 | 76.62 | 70.62 | **68.19** |
| *Implementation on bge-large-zh-v1.5* | | | | | | | | | | |
| InfoNCE | 45.14 | 75.86 | 88.19 | 67.33 | 76.01 | 59.51 | 84.92 | 76.86 | **71.73** | 58.09 |
| CoSENT | 39.66 | 68.22 | 85.68 | 63.79 | 67.28 | 55.90 | 82.36 | 69.49 | 66.55 | 64.70 |
| Mixed | 43.04 | 74.90 | 85.58 | 64.96 | 66.58 | 57.39 | 83.50 | 73.40 | 68.67 | 67.12 |
| CoDiEmb | 44.62 | 76.41 | 88.35 | 67.36 | 70.80 | 59.61 | 84.94 | 76.49 | 71.07 | **67.87** |

across several batch size configurations, using bge-large-zh-v1.5 as the backbone. The corresponding results are presented in Table 8. In the table, "IR Batch Size" refers to the number of queries, while the total number of documents involved in one iteration is this value multiplied by $(K^+ + K^-)$. In contrast, "STS Batch Size" denotes the number of standard text pairs.

As shown in Table 8, CoDiEmb maintains stable convergence even when the global batch sizes for IR and STS vary considerably. The overall score fluctuates by less than 0.5 points across all tested configurations, underscoring the framework's stability and robustness with respect to this critical hyperparameter.

Table 8: Robustness of CoDiEmb to different batch size configurations. The table shows performance as the per-device IR and STS batch sizes are varied. The total batch size is calculated over 32 GPUs. All experiments use the bge-large-zh-v1.5 backbone.

| IR Batch Size | STS Batch Size | Avg. IR | Avg. STS | Overall Score |
|---------------|----------------|---------|----------|---------------|
| $48 \times 32 = 1536$ | $28 \times 32 = 896$ | 70.99 | 67.78 | 138.77 |
| $52 \times 32 = 1664$ | $28 \times 32 = 896$ | 70.82 | 67.73 | 138.55 |
| $56 \times 32 = 1792$ | $32 \times 32 = 1024$ | 70.92 | 67.81 | 138.73 |
| $64 \times 32 = 2048$ | $32 \times 32 = 1024$ | 71.19 | 67.52 | 138.71 |
| $72 \times 32 = 2304$ | $32 \times 32 = 1024$ | 71.07 | 67.87 | **138.94** |

## A.6 Ablation Study of Loss Functions

Table 9: Ablation study of CoDiEmb's loss functions. The full model is compared against variants where each novel objective ($\mathcal{L}_{\text{RankKL}}$, $\mathcal{L}_{\text{PRO}}$, $\mathcal{L}_{\text{ENCE}}$) is individually removed. All experiments are conducted using the multilingual-e5-large backbone.

| Methods | Avg. IR | Avg. STS | Overall Score |
|---------|---------|----------|---------------|
| CoDiEmb (Full) | 70.62 | 68.19 | **138.81** |
| w/o $\mathcal{L}_{\text{RankKL}}$ | 70.26 | 67.50 | 137.76 |
| w/o $\mathcal{L}_{\text{PRO}}$ | 70.99 | 67.26 | 138.25 |
| w/o $\mathcal{L}_{\text{ENCE}}$ | 69.78 | 67.98 | 137.76 |

Building upon prior work, CoDiEmb introduces three novel objective functions to the training process of embedding models: (1) an extended contrastive loss with multiple positives and hard nega-

tives, denoted $\mathcal{L}_{\text{ENCE}}$; (2) a rank-normalized KL divergence loss, $\mathcal{L}_{\text{RankKL}}$; and (3) an adapted Preference Rank Optimization (PRO) loss, $\mathcal{L}_{\text{PRO}}$. As demonstrated in our main results, these objectives exhibit strong synergistic effects when adopted in concert.

To further ascertain whether each loss component contributes positively, we conduct an ablation study with multilingual-e5-large as the base encoder. The results are presented in Table 9. In this analysis, we systematically remove each component from the full model. Notably, the "w/o $\mathcal{L}_{\text{ENCE}}$" configuration reverts to a standard InfoNCE loss for the IR task, leaving other components intact.

The results clearly indicate that removing any of the proposed loss functions leads to a degradation in overall performance. The impact of $\mathcal{L}_{\text{RankKL}}$ and $\mathcal{L}_{\text{ENCE}}$ is particularly pronounced. Ablating either of these two components results in a performance drop of more than a full point on the overall score, an effect that could not be recovered even with careful hyperparameter retuning.

## A.7 COMPREHENSIVE PERFORMANCE ON THE FULL CMTEB BENCHMARK

In Appendix A.3, we provided a preliminary demonstration of CoDiEmb's transferability beyond the primary IR and STS domains by evaluating its performance on Pair Classification tasks using the `bge-large-zh-v1.5` (Xiao et al., 2024) backbone. To further substantiate the generality of our framework, this section extends CoDiEmb to the full CMTEB benchmark, which comprises a total of 31 tasks across six major categories. The detailed results are presented in Table 10.

We benchmark CoDiEmb against several leading embedding solutions, with performance metrics sourced directly from the official leaderboard. Due to computational constraints, we employed the 2B-scale MiniCPM (Hu et al., 2024) model as our backbone. Despite this, CoDiEmb surpasses models with significantly larger parameter counts and achieves SOTA-level performance.

In terms of implementation, we partitioned the entire training corpus into two categories based on whether the original data contains fine-grained labels. If a dataset possesses more than two label types (i.e., fine-grained labels), it is processed using the STS task methodology; otherwise, it is handled using the IR task methodology. For a detailed description of these two processing modes, please refer to Section 2.

The results in Table 10 demonstrate that CoDiEmb exhibits broad applicability and robust performance across a wide range of general-purpose embedding tasks. As indicated by the anonymous repository linked in the abstract, we commit to open-sourcing the complete training code to facilitate the development of more powerful encoder models by the community.

Table 10: Performance of CoDiEmb on the full CMTEB benchmark. The primary metric for each task category is reported.

| Methods | Backbone | IR | STS | Rerank | Cluster | CLS | PairCLS | Mean (Task) | Mean (Type) |
|---|---|---|---|---|---|---|---|---|---|
| Qwen3-Embedding | Qwen3-8B | 78.21 | 63.53 | 66.99 | **80.08** | 76.97 | 84.23 | 73.84 | 75.00 |
| E5-Mistral | Mistral-7B | 61.75 | 48.34 | 61.38 | 52.30 | 72.96 | 66.31 | 59.92 | 60.51 |
| GTE | Qwen2-7B | 75.70 | 65.20 | 69.24 | 66.06 | 75.77 | 81.16 | 71.62 | 72.19 |
| BGE | Gemma2-9B | 73.73 | 55.19 | 68.28 | 59.30 | 75.31 | 79.30 | 67.64 | 68.52 |
| CoDiEmb | MiniCPM-**2B** | **80.95** | **70.91** | **73.85** | 79.67 | **78.04** | **89.69** | **77.60** | **78.85** |

## A.8 HYPERPARAMETER CONFIGURATION AND TASK INSTRUCTION DESIGN

While the anonymous repository linked in the abstract provides comprehensive hyperparameter configurations for CoDiEmb across different backbone models, we summarize the critical settings here to facilitate reproducibility.

As formulated in Section 2.2, the objective function for STS tasks is a weighted sum of three components: $\mathcal{L}_{\text{STS}} = \alpha \mathcal{L}_{\text{Pearson}} + \beta \mathcal{L}_{\text{RankKL}} + \gamma \mathcal{L}_{\text{PRO}}$. With the exception of Appendix A.10, which specifically investigates the impact of loss weights on model performance, we set $\alpha = 2$, $\beta = 5$, and $\gamma = 0.5$ across all reported experiments. These weights were determined based on the relative magnitudes of the respective losses and remained fixed throughout the training process. We did not conduct an extensive hyperparameter search, as this configuration empirically proved sufficient for the model to significantly outperform baselines while maintaining stability across varying

batch sizes (Table 8). Regarding the temperature coefficients for $\mathcal{L}_{\text{RankKL}}$ and $\mathcal{L}_{\text{PRO}}$, we uniformly set both to $0.05$ for the `MiniCPM-Embedding` and `bge-large-zh-v1.5` backbones. For the `multilingual-e5-large` model, both were set to $0.02$.

For IR tasks, the number of positive samples per query ($K^+$) was set to 2, and the number of hard negatives ($K^-$) to 4 across all experiments. The choice of $K^+ = 2$ is motivated by the observation that it covers the relevant documents for the majority of queries (Figure 2). Conversely, $K^- = 4$ was selected to balance computational resources, as the effective pool of negatives scales by a factor of n_gpu $\times$ batch size when cross-device sampling is enabled. While determining the optimal $K^+$ and $K^-$ for each dataset via validation set tuning could potentially yield superior performance, we adopted a unified configuration in this work for the sake of efficiency.

It is worth noting that CoDiEmb allows for the flexible adjustment of hyperparameters for specific sample types through simple code modifications and the assignment of unique task identifiers $t$ at the data level (Section 2.1). This capacity for extensive customization represents a significant advantage of CoDiEmb over previous embedding training frameworks.

Regarding task instructions, we adhere to established conventions by employing symmetric instructions for STS data and asymmetric instructions for IR tasks (Su et al., 2023). To eliminate prompt engineering as a confounding variable and ensure a rigorous comparison, the instruction sets used for CoDiEmb are identical to those of the baselines in all experiments. The complete list of instructions is provided in Table 11.

Table 11: Instruction prompts used for C-MTEB. To ensure reproducibility, we present the original Chinese instructions alongside their English translations.

| Dataset | Original Instruction (Chinese) | English Translation |
|---|---|---|
| CmedqaRetrieval | 为医学问题生成表示，用于匹配相关医学文本 | Generate a representation for this medical question to retrieve relevant medical texts. |
| CovidRetrieval | 为新冠疫情相关问题生成表示，用于匹配相关文本 | Generate a representation for this COVID-19 related question to retrieve relevant texts. |
| DuRetrieval | 为用户日常搜索问题生成表示，用于匹配网页内容 | Generate a representation for this daily search query to retrieve relevant web pages. |
| MMarcoRetrieval | 为简短搜索问题生成表示，用于匹配相关长文本网页内容 | Generate a representation for this short search query to retrieve relevant long-text web content. |
| T2Retrieval | 为通用领域问题生成表示，用于匹配相关文本 | Generate a representation for this general domain question to retrieve relevant texts. |
| EcomRetrieval | 为用户购物搜索问题生成表示，用于匹配商品属性文本 | Generate a representation for this e-commerce search query to retrieve relevant product attribute texts. |
| MedicalRetrieval | 为患者症状描述文本生成表示，用于匹配诊疗方案或医学指南 | Generate a representation for this patient symptom description to retrieve relevant treatment plans or medical guidelines. |
| VideoRetrieval | 为简短用户问题生成表示，用于匹配视频标题或视频描述 | Generate a representation for this short user query to retrieve relevant video titles or descriptions. |
| AFQMC | 为金融领域句子生成表示 | Generate a representation for this financial domain sentence. |
| ATEC | 为金融领域客服场景的句子生成表示 | Generate a representation for this sentence from a financial customer service scenario. |
| BQ | 为银行客服对话中的句子生成表示 | Generate a representation for this sentence from a bank customer service dialogue. |
| LCQMC | 为通用领域句子生成表示 | Generate a representation for this general domain sentence. |
| PAWSX | 为英文翻译得到的中文句子生成表示 | Generate a representation for this Chinese sentence translated from English. |
| QBQTC | 为搜索引擎的搜索文本生成表示 | Generate a representation for this search engine query text. |
| STSB | 为简短的通用领域句子生成表示 | Generate a representation for this short general domain sentence. |

## A.9  GPU MEMORY USAGE AND TRAINING TIME

In this section, we compare the computational resource consumption of CoDiEmb against traditional approaches. Architecturally, CoDiEmb maintains full compatibility with mainstream pipelines. Specifically, the standard approach can be replicated within our framework by replacing the dynamic sampler with a single-dataset sampler using a fixed batch size and routing all samples to a unified processing branch.

To ensure parity in the total number of processed samples between the two settings, we employed the CoSENT loss for the baseline, as standard contrastive learning fails to fully leverage fine-grained annotation data. Furthermore, we aligned the batch size of the baseline with that of the IR component in CoDiEmb. We utilized `multilingual-e5-large` as the backbone and conducted distributed training on 16 GPUs, adhering to the same training and evaluation splits defined in Section 3.1. The corresponding results are presented in Table 12.

As evidenced by the results, CoDiEmb achieves comparable GPU memory usage and training duration to the standard approach, yet delivers significantly superior performance. This efficiency stems from two factors: first, CoDiEmb introduces no additional learnable parameters; second, the memory overhead of our ranking objective is markedly lower than that of contrastive learning, which necessitates solving a classification problem over a global batch—scaling with the number of GPUs during cross-device sampling—at every iteration. These findings confirm that CoDiEmb's "unified loading, distinct processing" paradigm is both computationally efficient and highly effective.

Table 12: Comparison of CoDiEmb and the standard method in terms of GPU memory usage, training time, and final performance. Consistent with Section 3.1, "Overall Score" represents the sum of the average IR and STS performance metrics.

| Methods | Memory Overhead | Training Time | IR Batch Size | STS Batch Size | Overall Score |
|---------|-----------------|---------------|---------------|----------------|---------------|
| CoDiEmb | 989.42 GB | 17 h 48 min | 64 | 48 | 138.81 |
| Standard | 997.24 GB | 17 h 40 min | 64 | 64 | 130.30 |

## A.10  THE IMPACT OF LOSS WEIGHTS ON MODEL PERFORMANCE

Table 13: Impact of varying loss weights on CoDiEmb performance. "Avg. STS" and "Avg. IR" denote the average scores across all STS and IR tasks in the CMTEB benchmark, respectively.

| $\alpha$ | $\beta$ | $\gamma$ | Avg. STS | Avg. IR | Overall Score |
|----------|---------|----------|----------|---------|---------------|
| 2 | 5 | 0.5 | 68.38 | 69.23 | 137.61 |
| 2 | 2.5 | 0.5 | 68.51 | 69.15 | 137.66 |
| 2 | 7.5 | 0.5 | 68.47 | 69.08 | 137.55 |
| 4 | 5 | 0.5 | 68.62 | 68.59 | 137.21 |
| 1 | 5 | 0.5 | 68.34 | 68.90 | 137.24 |
| 2 | 5 | 0.25 | 68.52 | 69.46 | **137.98** |
| 2 | 2.5 | 0.1 | 68.22 | 69.75 | 137.97 |

In this section, we investigate the sensitivity of CoDiEmb to variations in loss weight configurations. As detailed in Section 2.2, CoDiEmb optimizes IR tasks via an extended contrastive loss, while STS training is guided by a weighted combination of three ranking objectives. While this composite formulation ($\mathcal{L}_{\text{STS}} = \alpha\mathcal{L}_{\text{Pearson}} + \beta\mathcal{L}_{\text{RankKL}} + \gamma\mathcal{L}_{\text{PRO}}$) yields substantial performance gains, the introduction of three hyperparameters ($\alpha, \beta, \gamma$) necessitates an assessment of the model's robustness. To this end, we utilize the `multilingual-e5-large` backbone to evaluate performance variance across the 7 STS and 8 IR tasks of the CMTEB benchmark under varying weight assignments.

The experimental results are summarized in Table 13. We note that the baseline performance reported here (for $\alpha = 2$, $\beta = 5$, $\gamma = 0.5$) exhibits deviations from Table 1 due to adjustments in our computational infrastructure (specifically, GPU model and count) during the review process. However, the aggregate fluctuation remains within 1.5 points. This variance is primarily attributed

to the stochastic nature of our dynamic sampler, which initializes random seeds based on GPU rank, thereby altering the data consumption sequence during distributed training.

The results indicate that while specific weight values influence performance, CoDiEmb demonstrates strong overall robustness to hyperparameter variations. Furthermore, the analysis reveals that the default configuration employed in our main experiments was non-optimal, suggesting that CoDiEmb holds potential for further improvement through fine-grained hyperparameter tuning.

### A.11 LLM Usage Statement

We utilized large language models (LLMs) as a writing aid during the preparation of this manuscript. Specifically, LLMs were used for tasks such as improving grammar, refining phrasing, and ensuring clarity in the text. All core scientific contributions, including the methodological design, experimental setup, and analysis of results, were conceived and executed by the human authors.

