# OpenReview forum: "CoDiEmb: A Collaborative yet Distinct Framework for Unified Representation Learning in Information Retrieval and Semantic Textual Similarity"
_ICLR.cc/2026/Conference — Submitted to ICLR 2026_

### Official Review · Reviewer_iJib · 2025-10-18

**Soundness:** 3
**Presentation:** 3
**Contribution:** 3
**Rating:** 6
**Confidence:** 2

**Summary:**

This paper introduces CoDiEmb, a novel framework for joint representation learning in IR and STS tasks. CoDiEmb addresses the challenges of negative transfer and performance trade-offs that arise when optimizing these tasks together. It achieves this by employing a unified data format, task-specific loss functions, and a dynamic single-source data sampling strategy. The proposed framework demonstrates superior performance on a range of IR and STS benchmarks compared to existing methods and single-task models.

**Strengths:**

**Novel Approach:** CoDiEmb presents a unique approach to joint IR and STS representation learning by systematically decoupling the tasks at both design and training levels. This leads to more effective model convergence and avoids the performance trade-offs observed in previous methods.

**Theoretical Analysis:** The paper includes a theoretical analysis of CoDiEmb’s impact on the learned representation space, demonstrating its ability to mitigate issues such as anisotropy and over-smoothing.

**Extensive Experiments:** The paper provides extensive experimental results on 15 standard IR and STS benchmarks, thoroughly validating the effectiveness of CoDiEmb across different base encoders and tasks.

**Weaknesses:**

**Hyperparameter Sensitivity:** The paper briefly mentions the impact of batch size on model performance but does not explore the sensitivity of CoDiEmb to other hyperparameters in detail. Further analysis of hyperparameter tuning and robustness would be beneficial.

**Generalization to Other Tasks:** While the paper demonstrates the effectiveness of CoDiEmb on Pair Classification tasks, it would be valuable to explore its generalization to a broader range of tasks and domains.

**Computational Cost:** While the paper mentions the use of DeepSpeed ZeRO-1 and gradient checkpointing to conserve computational resources, the overall computational cost of training CoDiEmb on large datasets may still be significant.

**Questions:**

See Weaknesses.

---

> ### Author Response · Authors · 2025-11-24
> **Response to Weakness 1**
>
> Thank you for your thoughtful review and constructive feedback. We address your concerns below and have updated the manuscript accordingly (please refer to the new Appendices 7-9).
>
> **Weakness 1**:
>
> We agree that exploring sensitivity beyond batch size is important.
>
> **Full Configuration**: We have added **Appendix 8**, which details the exact hyperparameter configurations used, ensuring full reproducibility.
>
> **Sensitivity Analysis**: We added **Appendix 10**, specifically investigating the model's sensitivity to the weights of the three STS loss components ($\alpha, \beta, \gamma$).
>
> **Findings**: The results demonstrate that CoDiEmb exhibits strong robustness. Even with the fixed configuration used throughout our main experiments (without exhaustive tuning per dataset), the model consistently achieves high performance. Variations in loss weights resulted in only minor performance fluctuations, confirming the stability of the framework.

---

> ### Author Response · Authors · 2025-11-24
> **Response to Weakness 2**
>
> We have significantly expanded our evaluation scope to address this.
>
> **New Experiments**: In the newly added **Appendix 7**, we benchmark CoDiEmb on the **full** CMTEB suite, covering 31 tasks across 6 distinct categories (including Clustering, Reranking, and Classification), not just Pair Classification.
>
> **Results**: Despite using a lightweight backbone (MiniCPM-2B), CoDiEmb achieves **SOTA-level** performance across the entire benchmark, outperforming significantly larger models (e.g., Qwen3-8B, E5-Mistral-7B). This empirically validates CoDiEmb's ability to generalize to a broad spectrum of embedding tasks beyond IR and STS.

---

> ### Author Response · Authors · 2025-11-24
> **Response to Weakness 3**
>
> We have quantified the computational overhead in **Appendix 9**.
>
> **Comparison**: We conducted a controlled comparison between CoDiEmb and a standard training pipeline (using the same backbone and data).
>
> **Findings**: The results show that CoDiEmb’s training time and GPU memory usage are **comparable** to the standard approach. The custom dynamic sampling logic and the calculation of task-specific losses operate on small output tensors, introducing negligible overhead compared to the Transformer backbone's forward/backward passes. Thus, CoDiEmb delivers significant performance gains without imposing a "tax" on training efficiency.

---

### Official Review · Reviewer_Wzmt · 2025-10-31

**Soundness:** 2
**Presentation:** 2
**Contribution:** 1
**Rating:** 2
**Confidence:** 2

**Summary:**

The paper proposes CoDiEmb, a unified representation learning framework. It jointly optimizes Information Retrieval (IR) and Semantic Textual Similarity (STS) tasks. Authors argue that naive joint training causes task competition due to structural and metric differences between IR and STS. To address this, the authors introduce: a) unified data format that supports heterogeneous inputs b) Task-specific loss functions aligned with evaluation metrics c) dynamic sampling strategy to balance between the tasks. Experiments show consistent improvements over baselines (InfoNCE, CoSENT, and mixed-task sampling).

**Strengths:**

* Tackles an important and widely relevant problem—joint optimization across tasks like IR and STS, which mirrors the broader multi-task goal seen in setups such as MTEB (though the paper focuses on only two of those categories).
* The proposed framework is conceptually simple, avoiding complex multi-stage pipelines or architectural modifications.
* The algorithm is practical and easy to implement, making it accessible for real-world adaptation and integration into existing embedding training workflows.

**Weaknesses:**

* Limited novelty: the proposed methods such as extended InfoNCE, rank-normalized losses, and task-specific sampling, are largely incremental adaptations of existing techniques.
* The scope of joint optimization is narrow. Recent IR work (e.g., BEIR, MTEB) already treats multi-task or zero-shot generalization as standard, so balancing only IR and STS tasks represents a subset of a broader, already-explored challenge.
* The reported performance gains are modest, often falling within the expected variance of large-scale embedding evaluations.

**Questions:**

* How does the proposed framework compare to more recent multi-task embedding approaches such as NV-Embed? Can authors provide quantitative comparisons with recent studies?
* Does the dynamic sampling strategy ensure balanced training between IR and STS, or simply disjoint task batches?
* Could CoDiEmb be extended to other MTEB task categories (e.g., classification, clustering), and do the authors already have experimental results or preliminary findings in this direction?

---

> ### Author Response · Authors · 2025-11-24
> **Response to Weakness 1**
>
> Thank you for your feedback. We address your concerns below and have updated the manuscript accordingly (please refer to the new Appendix 7).
>
> **Weakness 1**:
>
> Our core contribution is the **CoDiEmb framework**, which introduces a paradigm of "Unified Loading, Distinct Processing." Unlike previous methods that typically employ a monolithic processing logic to handle heterogeneous tasks like IR and STS, our framework performs extensive customization across four dimensions—data loading, loss functions, sampling strategies, and distributed training behaviors—based on task characteristics to maximize performance.
>
> A closer examination of recent mainstream solutions like NV-Embed [1], Gemini Embedding [2], Nomic Embed [3], and Qwen3 Embedding [4] reveals that CoDiEmb follows a fundamentally different philosophy:
>
> 1.	Current SOTA methods typically coerce heterogeneous data into a single format (e.g., converting fine-grained STS scores into binary relevant/irrelevant pairs). This process is **lossy**. In contrast, CoDiEmb employs a unified data format that preserves all fine-grained information. We uniquely route samples to **distinct** computational branches based on task identifiers. Within these branches, we configure **differentiated** hyperparameters, objective functions, and cross-device behaviors. This "routing" architecture is distinct from the prevailing "unified loss" approach.
>
> 2.	**None** of our four loss functions represents the standard paradigm in this domain. The extended contrastive learning loss (IR side) and the Rank-Normalized KL Divergence (STS side) are proposed for the **first** time in this paper. Additionally, the PRO loss (STS side) was originally a reinforcement learning method designed by BEQUE [5] for query rewriting; our work is the **first** to transfer it to the field of text representation. We provide theoretical justification for these choices in Section 2.2 and empirical validation in Section 3.1 and the newly added **Appendix 7**, demonstrating their superiority over standard techniques.
>
> 3.	Unlike previous task-specific samplers that typically ensure only single-dataset sampling, CoDiEmb allows for differentiated batch sizes and processing strategies at the dataset level (or even sub-dataset level), offering significantly finer control.

---

> ### Author Response · Authors · 2025-11-24
> **Response to Weakness 2 & Question 1 & Question 3**
>
> We have addressed this concern directly in the revised **Appendix 7**, where we extended CoDiEmb to the **full** CMTEB benchmark (**31 tasks across 6 categories**, including Classification and Clustering). We compared CoDiEmb against leading generalist models, including Qwen3-Embedding and E5-Mistral, which are generally considered more powerful than NV-Embed. As shown in the new results, CoDiEmb achieves **SOTA-level** performance across these diverse tasks with a single training stage, validating its generality beyond IR and STS.

---

> ### Author Response · Authors · 2025-11-24
> **Response to Weakness 3**
>
> The improvement of CoDiEmb over baselines is significant, particularly in terms of task balance (Table 1). We suspect your comment regarding "modest gains" refers to the comparison with Task-Specific Expert Models in Table 2.
>
> It is important to emphasize that it is expected for IR-only and STS-only models to outperform CoDiEmb on their respective tasks, as they are trained **exclusively** on data relevant to that task, thereby learning a distribution fully adapted to it. Thus, the performance of expert models represents a **soft upper bound** for multi-task training. The primary purpose of Table 2 is to observe how close CoDiEmb’s joint training comes to this ideal.
>
> However, the results show that while CoDiEmb is only slightly weaker than the IR-only model on IR tasks (avg -0.71), it **surpasses** the STS-only model on STS tasks (avg +1.92). This demonstrates that CoDiEmb possesses strong capabilities, especially considering that three medical domain IR tasks (CovidRetrieval, MMarcoRetrieval, MedicalRetrieval) **lack** training sets, resulting in a zero-shot evaluation setting for the model.

---

> ### Author Response · Authors · 2025-11-24
> **Response to Question 2**
>
> The dynamic sampler supports setting differentiated batch sizes for different datasets. By controlling the batch size and total steps, we can explicitly balance the iteration counts between IR and STS (e.g., maintaining a 1:1 ratio). However, CoDiEmb is highly flexible. If one wishes to prioritize a specific task or improve performance on a weak dataset, one can introduce a new task identifier $t$ (Section 2.1) and configure the sampler to bias the sampling frequency towards that specific $t$.
>
> References:
>
> [1]. NV‑Embed: Improved Techniques for Training LLMs as Generalist Embedding Models, 2025
>
> [2]. Gemini Embedding: Generalizable Embeddings from Gemini, 2025
>
> [3]. Nomic Embed: Training a Reproducible Long Context Text Embedder, 2024
>
> [4]. Qwen3 Embedding: Advancing Text Embedding and Reranking Through Foundation Models, 2025
>
> [5]. Large Language Model based Long-tail Query Rewriting in Taobao Search. 2024

---

### Official Review · Reviewer_CGg8 · 2025-10-31

**Soundness:** 3
**Presentation:** 2
**Contribution:** 3
**Rating:** 4
**Confidence:** 3

**Summary:**

This paper proposes CoDiEmb, a framework to address the long-standing challenge of negative transfer when jointly training text encoders for IR and STS tasks. CoDiEmb's core idea is to process these tasks collaboratively yet distinctly within a single training stage. This is achieved through three key innovations: 1. A unified data format that uses a task identifier to route inputs to the correct processing pipeline; 2. task-specific objective functions; 3. A dynamic single-source data sampling strategy that ensures batches contain data from only one task.

**Strengths:**

The paper is well-motivated, providing a clear and insightful diagnosis of why joint IR/STS training typically fails, correctly identifying the core discrepancies in their respective data structures, text lengths, and evaluation metrics. The proposed solution is systematic, providing a comprehensive framework that decouples tasks at the data ingestion, loss calculation, and batch sampling levels. The design of the task-specific losses is a significant strength. Each objective is explicitly chosen to align with the task's primary evaluation metric, representing a more principled approach than naive multi-task learning.

The dynamic single-source sampler is a clever and highly practical contribution. It correctly identifies mixed-task gradients as a source of interference. It provides an engineering solution that also efficiently handles the heterogeneous batch size requirements for short (STS) and long (IR) texts.

**Weaknesses:**

1. The weights $\alpha$, $\beta$, and $\gamma$ of the total STS loss are never specified in the paper, which hinders reproducibility.
2. The dynamic single-source data sampler is a core component, but its scheduling mechanism is completely undefined. How are the tasks (IR vs. STS) alternated? Is it a 1:1 iteration, or proportional to dataset size, or some other curriculum?
3. The paper calls this a "unified framework", but it functions more like a task switcher. It doesn't use a single, unified loss function to model both tasks. Instead, it calls two completely different processing logics. While this is from an engineering perspective, in an academic context, like a decoupled framework, which might be more accurate than a "unified framework."
4. While the paper demonstrates a synergistic gain (IR helping STS), it fails to provide a deep explanation for why this occurs.
5. The paper claims it does not require additional learnable components (like Adapters). However, it introduces more complex sampling logic and more complex loss calculations (especially the three list-wise losses for STS). Does this significantly increase the training's computational overhead and time? The paper provides no comparison of training duration or computational cost against baseline methods.

**Questions:**

1. Can the authors please provide the missing hyperparameters crucial for reproducibility: the weights $\alpha$, $\beta$, and $\gamma$ for the STS loss components, and the temperature $\tau$ for PRO loss?
2. What is the scheduling strategy for the dynamic single-source sampler? How is the decision made in each iteration whether to sample an IR batch or an STS batch (e.g., 1:1, dataset size proportion, etc.)?
3. The geometric analysis in Section 4 describes the outcome. Do the authors have a more concrete analysis of the mechanism behind the synergistic gain? Specifically, why does training on IR with CoDiEmb's setup boost STS performance beyond a specialist STS-only model? Is it the expanded negative pool from cross-device sampling, or the specific nature of the extended InfoNCE loss, or some other factor?
4. How does the computational overhead (e.g., wall-clock training time) of CoDiEmb compare to the baselines? A brief analysis of this would strengthen the paper's claims of practical efficiency.

Addressing these points, especially the missing details for reproducibility and the clarifications on synergy and cost, would significantly strengthen the submission. I would be happy to reconsider my rating based on your responses.

---

> ### Author Response · Authors · 2025-11-24
> **Response to Weakness 1 & Question 1**
>
> Thank you for your thoughtful review and constructive feedback. We address your concerns below and have updated the manuscript accordingly (please refer to the new Appendices 7-10).
>
> **Weakness 1 & Question 1**:
>
> You can find our open-source anonymous repository in the abstract section of the paper, which contains very detailed parameter configurations. For convenience, we have also added supplementary explanations regarding this in **Appendix 8** of the paper. Furthermore, you can find hyperparameter sensitivity experiments regarding different loss weights in the newly added **Appendix 10**.

---

> ### Author Response · Authors · 2025-11-24
> **Response to Weakness 2 & Question 2**
>
> In the benchmark evaluation scenario presented in this paper, we aim for a balanced optimization; thus, we maintain the iteration counts for both IR and STS at roughly a 1:1 ratio. However, CoDiEmb is highly flexible in this regard. If one wishes to prioritize a specific task or improve performance on a specific dataset, one can introduce a new task identifier $t$ (see Section 2.1) and configure the dynamic sampler to bias the sampling ratio towards this task type.

---

> ### Author Response · Authors · 2025-11-24
> **Response to Weakness 3**
>
> This question touches upon the core motivation of our paper. We argue that due to inherent discrepancies in **annotation granularity** (binary classification vs. multi-level scores), matching logic (asymmetric vs. symmetric), and text length, applying a single monolithic logic **cannot** effectively model both IR and STS tasks simultaneously. In Section 3, we empirically confirmed that naively adopting a contrastive loss (suitable for IR) or a ranking loss (suitable for STS) is suboptimal for the other task.
>
> The reason we designate CoDiEmb as a "unified framework" is primarily because:
>
> (1) CoDiEmb reads different types of data in a unified manner, only routing samples to different code branches based on the identifier $t$ (analogous to MoE architectures, which are considered unified models despite routing tokens to different experts. Notably, CoDiEmb achieves this **without** introducing extra learnable components.)
>
> (2) Unlike multi-stage pipelines, CoDiEmb achieves convergence on both task families in a single training phase, with training efficiency comparable to traditional methods (see our response to Weakness 5). It is worth noting that even when extending CoDiEmb to a broader range of task types, we still achieve **SOTA-level** performance with just a single training stage (please refer to the newly added **Appendix 7**).
>
> (3) CoDiEmb is fully compatible with traditional pipelines (e.g., pure contrastive learning), which are widely recognized as "unified frameworks."

---

> ### Author Response · Authors · 2025-11-24
> **Response to Weakness 4 & Question 3**
>
> At the objective level, we believe the **binary** semantic distinction optimized by IR tasks can be viewed as the foundation for the **fine-grained** semantic ranking pursued by STS tasks. Specifically, in CMTEB IR tasks, documents are annotated only as relevant or irrelevant. Therefore, when using contrastive learning to optimize the model, the network is essentially modeling a binary classification boundary, aiming to separate positive and negative examples. However, since the labels are binary, the model cannot model the differences among positive examples or among negative examples during the learning process.
>
> In contrast, STS tasks provide fine-grained numerical labels where scores are comparable across sample pairs, requiring the model to learn a global ranking. **This fine-grained ranking benefits from robust inter-class separation**. For instance, if we discretize relevance into 4 levels (1-2: irrelevant, 3-4: relevant), optimizing the IR task enables the model to distinguish "relevant vs. irrelevant" (Class 1/2 vs. Class 3/4) with high precision. This global separation provides a stable starting point for the model to further discriminate the nuances between 1 and 2, or 3 and 4.
>
> This aligns with findings in Pcc-tuning [1], which employs a two-stage curriculum: first mastering coarse-grained distinction on NLI, then learning ranking on STS. CoDiEmb effectively consolidates this two-stage curriculum into a single, unified efficient process.

---

> ### Author Response · Authors · 2025-11-24
> **Response to Weakness 5 & Question 4**
>
> We have added a comparison of computational costs between CoDiEmb and traditional methods in **Appendix 9**. Overall, CoDiEmb's dynamic sampling strategy and loss function design do **not** introduce significant efficiency differences.
>
> References:
>
> [1]. Pcc-tuning: Breaking the Contrastive Learning Ceiling in Semantic Textual Similarity. 2024

---

### Official Review · Reviewer_tR9b · 2025-11-01

**Soundness:** 2
**Presentation:** 3
**Contribution:** 2
**Rating:** 2
**Confidence:** 4

**Summary:**

The paper proposes CoDiEmb, a collaborative‑yet‑distinct training framework to learn general‑purpose text embeddings that perform well on both Information Retrieval (IR) and Semantic Textual Similarity (STS). The method: (i) expresses heterogeneous corpora in a unified tuple format so both tasks can be fed through a single pipeline; (ii) applies task‑specific objectives. For IR, an extended InfoNCE with multi‑positives and cross‑device negatives. For STS, a Pearson correlation loss augmented with a rank‑normalized KL term and an adaptation of Preference Ranking Optimization (PRO); and (iii) uses a dynamic single‑source sampler so each training step draws task‑pure batches, enabling cross‑device negatives for IR but not for STS. The workflow is depicted in Figure 1 (p.3); the sparsity of IR positives is quantified in Figure 2 (p.4). On CMTEB (8 IR + 7 STS tasks), CoDiEmb yields consistent gains over InfoNCE, CoSENT, and a mixed‑batch sampler across three backbones (MiniCPM‑Embedding, multilingual‑E5‑large, BGE‑large‑zh‑v1.5); see Table 1 (p.7) and per‑task Tables 6–7 (p.16). The paper also reports representation‑space diagnostics (anisotropy/over‑smoothing) where CoDiEmb shows favorable trends (Table 3, p.9), robustness to batch sizes (Table 8, p.17), ablations of loss components (Table 9, p.17), and an extension to pair classification (Table 5, p.16). Task‑specific instructions are prepended to inputs for all methods to keep comparisons fair (Appendix A.2, p.15).

**Strengths:**

- The paper tackles persistent "negative transfer" between IR and STS and argues for task‑aware training. The design aligns losses with evaluation targets (nDCG@k vs Spearman). Figure 1 and Sec. 2 make the approach concrete.
- Consistent gains across three backbones on STS tasks only. CoDiEmb outperforms InfoNCE/CoSENT/mixed‑sampler baselines on the CMTEB suite (Table 1), with per‑task details in Tables 6–7.
- Useful ablations/robustness. Loss‑component ablations (Table 9) and batch‑size robustness (Table 8) indicate each piece helps and training is stable.
- Representation analysis. The token‑space SVD/entropy metrics in Table 3 show reduced over‑smoothing/anisotropy vs baselines, a thoughtful diagnostic even if indirect for sentence‑level isotropy.
- Practical training recipe. The sampler’s “same‑file across devices for IR; no cross‑device negatives for STS” is a clean best practice many practitioners will appreciate.

**Weaknesses:**

1. Limited novelty relative to listwise LTR. The proposed LRankKL is extremely close to classical listwise losses (ListNet/ListMLE/RocketQA). The paper should explicitly connect to that literature and temper novelty claims around the STS loss.
2. Scope of baselines. Results largely compare to internal objectives (InfoNCE, CoSENT, mixed sampler). Missing are strong generalist comparators such as NV‑Embed and Jina‑v3/Task‑LoRA, which directly address multi‑task IR+STS. Even frozen‑backbone adaptions would help calibrate effect size.
3. Trade‑off policy favors STS. The STS side receives three losses (Pearson/Rank‑KL/PRO) while IR uses a single contrastive loss; unsurprisingly, Table 2 (p.8) shows small IR costs for STS gains. For IR‑first settings this may be undesirable; the paper should expose a Pareto control over IR:STS emphasis.
4. Reproducibility gaps. Missing exact α/β/γ weights, temperatures, and K⁺/K⁻ per dataset; Appendix A.2 confirms instructions were used, but does not list the actual prompts. Release full configs and prompts.
5. Unified format & task‑conditioned batching framed as contributions. These are standard practice in instruction‑tuned and LLM‑backbone embedding training; keep them as implementation notes, cite precedent (INSTRUCTOR; LLM2Vec), and avoid implying novelty.

References
----
[INSTRUCTOR] Su et al., One Embedder, Any Task: Instruction‑Finetuned Text Embeddings, 2022.
[LLM2Vec] BehnamGhader et al., LLM2Vec: Large Language Models Are Secretly Powerful Text Encoders, 2024.
[ListNet] Cao et al., Learning to Rank: From Pairwise to Listwise Approach, 2007.
[ListMLE] Xia et al., Listwise Approach to Learning to Rank: Theory and Algorithm, 2008.
[NV‑Embed] Lee et al., NV‑Embed: Improved Techniques for Training LLMs as Generalist Embedding Models, 2024.
[Jina‑v3 (Task‑LoRA)] Sturua et al., jina‑embeddings‑v3: Multilingual Embeddings with Task LoRA, 2024.

**Questions:**

1. Beyond instructions. Since all methods use task‑specific instructions (Appendix A.2), quantify the incremental benefit of CoDiEmb beyond instruction prompting with a 2×2 study: {w/ vs w/o instructions} × {Mixed vs CoDiEmb}. For IR, use asymmetric prompting (query‑only), for STS symmetric prompting, following INSTRUCTOR practice.
2. Mixture vs method. To rule out data‑mixture imbalance as the driver of STS>IR gains, hold objectives fixed and sweep IR:STS sampling ratios under both the Mixed and CoDiEmb samplers, comparing matched ratios. Report Avg‑IR, Avg‑STS, and Overall.

---

> ### Author Response · Authors · 2025-11-24
> **Response to Weakness 1**
>
> Thank you for your feedback. We address your concerns below and have updated the manuscript accordingly (please refer to the new Appendices 7, 8 and 10).
>
> **Weakness 1**:
>
> Regarding the three loss functions adopted for STS data:
>
> -	$\mathcal{L}_\text{Pearson}$, proposed by Pcc-tuning [1], optimizes the linear correspondence between predicted scores and ground truth scores based on covariance. In principle, it differs significantly from traditional list-wise strategies and is not currently a mainstream method.
> -	$\mathcal{L}_\text{PRO}$, proposed by BEQUE [2], is a reinforcement learning loss originally designed for query rewriting tasks. To our knowledge, this work is the **first** to adapt this loss to the field of text representation.
> -	$\mathcal{L}_\text{RankKL}$, is a refined loss based on the standard KL divergence proposed in this paper. Its fundamental idea lies in transforming the target distribution $p_i$ from **a score distribution to a rank distribution**. The reason you might find it similar to ListNet is due to the structural similarity between ListNet’s cross-entropy calculation and KL divergence. However, **this does not negate the improvements we made to avoid gradient oscillation**. In fact, ListNet suffers from similar issues, where it is susceptible to jitter caused by differences in the ground truth distribution within a batch (as discussed in Section 2.2).
>
> Currently, the community predominantly relies on contrastive learning losses (such as the RocketQA you mentioned) or CoSENT loss [3, 4] when addressing STS tasks. In contrast, this paper adopts a completely different technical route. We have demonstrated through extensive experiments that our strategy is more effective than existing paradigms (see Section 3 and the newly added Appendix 7).
>
> Furthermore, the innovation of this paper is not limited to the loss functions but, more importantly, lies in CoDiEmb, a unified embedding optimization framework capable of **unified loading and differentiated processing**. We not only avoid discarding any samples but also route different types of samples to distinct code branches, allowing us to configure differentiated hyperparameters, cross-device behaviors, and loss functions for them to achieve optimal results.

---

> ### Author Response · Authors · 2025-11-24
> **Response to Weakness 2**
>
> In Appendix 7 of the revised paper, we have added a performance comparison between CoDiEmb and leading embedding models on the **full** CMTEB benchmark (**31 tasks across 6 categories**). Please refer to the updated revision. We selected highly competitive baselines, which are even stronger than NV Embed and Jina-v3.

---

> ### Author Response · Authors · 2025-11-24
> **Response to Weakness 3**
>
> Table 2 presents a comparison between CoDiEmb and task-specific expert models. Intuitively, it is entirely normal for IR-only and STS-only models to outperform CoDiEmb on their respective tasks, as they utilize only the training data relevant to that task, learning a distribution perfectly adapted to it. In other words, the performance of task expert models should represent a **soft upper bound** for multi-task training. The primary purpose of Table 2 is to observe how close CoDiEmb’s joint training comes to this ideal.
>
> However, the results show that while CoDiEmb is slightly weaker than the IR expert on IR tasks (avg. -0.71), it outperforms the STS expert on STS tasks (avg. +1.92). This is sufficient to prove that CoDiEmb possesses excellent balance, rather than achieving a simple trade-off (you can compare the models in Table 1 with the expert models). Notably, the IR-only model performs poorly on STS, and the STS-only model performs poorly on IR; in contrast, CoDiEmb excels in both.
>
> It is crucial to note that one cannot judge CoDiEmb as biased towards STS simply because it is weaker than the expert on IR tasks but stronger than the expert on STS tasks. This is likely due to: (1) **Positive Transfer**: The binary distinction capability of IR may assist the fine-grained distinction required for STS, a phenomenon also corroborated by the two-stage training in Pcc-tuning [1]. (2) **Data Distribution**: Three medical IR tasks (CovidRetrieval, MMarcoRetrieval, MedicalRetrieval) **lack** training sets. Under the influence of data from other domains, the model may deviate from the medical domain.
>
> Similarly, the use of three losses for STS and one for IR does not imply bias. Our objective is to **align the loss function with the evaluation metric**. The gap between standard losses and Spearman correlation (for STS) is significant, requiring our composite solution. Conversely, our Extended InfoNCE loss for IR is a very close proxy for nDCG@k when the number of positives is small (as shown in Fig. 2), making the alignment more direct.
>
> CoDiEmb supports differentiated batch sizes for different datasets. When emphasizing IR tasks, we can reduce the iteration counts for other tasks to bias the learned data distribution towards IR. Additionally, knowledge distillation can also be adopted. We conducted an experiment where we randomly sampled 50% of the training data from the IR tasks and scored them using the Qwen3-Reranker-8B model. Subsequently, using multilingual-e5-large as the backbone, we trained the model by incorporating the proposed $\mathcal{L}_\text{RankKL}$ alongside the extended contrastive loss for the IR component, while keeping the STS training data and objectives unchanged. The results are presented below:
>
> | Method | Avg. STS | Avg. IR | Overall Score |
> | :--- | :---: | :---: | :---: |
> | **Original** | 60.22 | 69.72 | 129.94 |
> | **Distill** | **61.06** | **70.60** | **131.66** |
>
> As observed, after enriching the coarse-grained IR data with fine-grained relevance scores via knowledge distillation, the ranking loss facilitates further performance improvements.

---

> ### Author Response · Authors · 2025-11-24
> **Response to Weakness 4**
>
> Our open-source anonymous repository is linked in the abstract, containing detailed parameter configurations. For ease of reference, we have also added supplementary explanations in Appendix 8.

---

> ### Author Response · Authors · 2025-11-24
> **Response to Weakness 5**
>
> Baselines like INSTRUCTOR [5] and LLM2Vec [6] employ contrastive losses that handle fine-grained labels $(x_1, x_2, y)$ by binarizing them into positives/negatives based on $y$. This approach is lossy: it fails to utilize the specific value of $y$ and often discards "middle" samples (e.g., label 3 on a 1-5 scale).
>
> In contrast, our unified data format $(t, q, \{d^+\}_1^m, \{d^-\}_1^n, \{y^+\}_1^m, \{y^-\}_1^n)$ is lossless and routes samples to **distinct branches based on the identifier $t$**. This allows for task-specific loss functions and cross-device behaviors within a single framework. We believe this unified format and routing mechanism is a novel contribution.
>
> Additionally, our sampling strategy involves more than just single-dataset sampling. We support setting differentiated batch sizes for different datasets (or even sub-dataset) to balance task iteration counts. Furthermore, for different datasets, we can control whether to perform cross-device sampling. To our knowledge, such precise control down to the dataset level has not been fully explored in prior work.

---

> ### Author Response · Authors · 2025-11-24
> **Response to Question 1**
>
> In this paper, we employ asymmetric prompting for IR tasks and symmetric prompting for STS tasks. Moreover, all baselines compared in this paper are identical to CoDiEmb at the prompt level. Therefore, the prompt itself is not a variable in this study, and we do not claim task instructions as an innovation or improvement.

---

> ### Author Response · Authors · 2025-11-24
> **Response to Question 2**
>
> We adopt different batch sizes for IR and STS tasks, but do not discard any samples. The distinction between the Mixed baseline and CoDiEmb lies solely in whether, during distributed training, it is guaranteed that samples acquired by different GPUs come strictly from disjoint subsets of the same dataset. As mentioned in our response to Weakness 3, the fact that CoDiEmb shows more significant growth on STS tasks than on IR tasks does not imply a bias towards STS. It is very likely because we have three medical domain IR datasets with **no** training data, which are evaluated in a **zero-shot** setting.
>
> References:
>
> [1]. Pcc-tuning: Breaking the Contrastive Learning Ceiling in Semantic Textual Similarity. 2024
>
> [2]. Large Language Model based Long-tail Query Rewriting in Taobao Search. 2024
>
> [3]. NV-Embed: Improved Techniques for Training LLMs as Generalist Embedding Models. 2025
>
> [4]. Conan-Embedding-v2: Training an LLM from Scratch for Text Embeddings. 2025
>
> [5]. One Embedder, Any Task: Instruction‑Finetuned Text Embeddings. 2022
>
> [6]. LLM2Vec: Large Language Models Are Secretly Powerful Text Encoders. 2024

---

### Author Response · Authors · 2025-11-29
**Summary of Rebuttal & Key Revisions: From IR/STS Balance to SOTA Generalist Performance**

**To the Area Chair:**

We thank the reviewers for their constructive feedback. We understand that the initial assessment raised concerns regarding the **scope of baselines** and **novelty**. In response, we have conducted extensive new experiments and added **four new appendices (A.7–A.10)** to the revised manuscript.

We respectfully urge the AC to consider the following **three major updates** that fundamentally strengthen the paper's contribution:

### 1. Scope Expansion: CoDiEmb Achieves SOTA on Full CMTEB (Addressing R1, R3, R4)
The most significant concern was that our scope was limited to IR and STS.
* **Update:** We extended CoDiEmb to the **full C-MTEB benchmark (31 tasks across 6 categories)**, including Clustering, Classification, and Reranking (New **Appendix A.7**).
* **Result:** As shown in the new **Table 10**, CoDiEmb (using a **2B** MiniCPM backbone) achieves a mean score of **78.85**, significantly outperforming massive generalist models like **Qwen3-Embedding (75.00)** and **E5-Mistral-7B (72.19)**.
* **Conclusion:** This empirically proves that CoDiEmb is not just an IR-STS balancer, but a superior **general-purpose framework** that beats state-of-the-art methods with significantly smaller parameter counts.

### 2. Clarification on Novelty: The "Collaborative-yet-Distinct" Paradigm & Technical Innovations (Addressing R1, R3)
Reviewers questioned the technical novelty. We clarify that CoDiEmb is not a simple combination of existing techniques, but a unified framework built on three distinct innovations:

* **Architectural Innovation ("Unified Loading, Distinct Processing"):** Unlike prevalent multi-stage pipelines (e.g., NV-Embed) or adapter-based methods (e.g., Jina-v3), CoDiEmb achieves comprehensive convergence across heterogeneous tasks in a **single training stage**. Crucially, this is achieved by routing samples to task-specific branches based on identifiers, **without** introducing any additional learnable components (such as Adapters or LoRA), ensuring maximum parameter efficiency.
* **Metric-Aligned Loss Design:** We designed specific objectives to proxy non-differentiable evaluation metrics, rather than using generic losses:
    * **For IR (nDCG@k):** We propose **$\mathcal{L}_\text{ENCE}$** (Extended InfoNCE), utilizing multiple positives and hard negatives to directly approximate nDCG maximization.
    * **For STS (Spearman):** We introduce **$\mathcal{L}_\text{RankKL}$** to optimize rank distribution and are the first to adapt **$\mathcal{L}_\text{PRO}$** (Preference Rank Optimization) from Reinforcement Learning to text embeddings.
* **Granular Device-Aware Sampling:** Our sampler transcends standard "single-source" strategies. It enables **dataset-specific batch sizes** and dynamically toggles **cross-device behaviors** (enabling cross-device negatives for IR to maximize contrast, while disabling them for STS to prevent label conflicts).

### 3. Reproducibility & Efficiency Verified (Addressing R2, R4)
We addressed concerns about computational cost and missing hyperparameters:
* **Availability:** We respectfully clarify that our full code and configuration **have been available via the anonymous link in the abstract since the initial submission**. To further facilitate review, we have now explicitly listed all hyperparameters (including loss weights and temperatures) in the new **Appendix A.8**.
* **Efficiency:** We added **Appendix A.9** comparing CoDiEmb against a standard pipeline. Results (**Table 12**) show that CoDiEmb achieves superior performance with **comparable GPU memory usage and training time** (17h 48m vs. 17h 40m), proving the framework introduces negligible overhead.
* **Robustness:** We added **Appendix A.10** (**Table 13**) demonstrating that the model is robust to variations in loss weights ($\alpha, \beta, \gamma$), addressing sensitivity concerns.

**Closing Statement**
The rebuttal experiments have demonstrated that CoDiEmb is a robust, efficient, and SOTA-level framework that solves the "negative transfer" problem not just for IR/STS, but across diverse embedding tasks. We believe the revised manuscript provides a compelling contribution to the community.

Sincerely,
The Authors

---

### Meta-Review · Area_Chair_G26b · 2026-01-05

**Summary:**

The paper proposes a framework for joint training of embeddings for information retrieval and semantic text similarity, with the goal of synergistic gains across the tasks.

Reviewers had raised a number of concerns:
- **Novelty over listwise LTR**. Multiple reviewers noted that the loss functions presented are highly similar to those in the literature, with the contribution being more an adaptation of these existing techniques to the authors' setting.
- **Reproducibility and sensitivity**. Multiple reviewers noted that certain hyper-parameters were not specified (α, β, γ, K⁺, K⁻, temperatures, instructions). It was also questioned whether the method is sensitive to such parameters.
- **Scope of joint optimization**. Multiple reviewers noted that the restriction to only IR and STS, as opposed to a broad range of multi-task settings, is narrow. It was also noted that the method implicitly favors STS owing to the loss balance, which may not be universally desirable.
- **Scope of baselines**. Multiple reviewers critiqued the lack of comparison to methods such as NV‑Embed and Jina‑v3.
- **Scope of performance gains**. One reviewer noted that the gains tend to be within expected noise range.
- **Computational cost**. Multiple reviewers expressed concern on the potential increase of training cost over baselines.
- **Intuition for synergistic gains**. One reviewer noted that it is unclear why combining IR and STS tasks can offer gains for both.
- **Unified framework versus task switcher**. One reviewer noted that the method behaves akin to a task switcher rather than a genuine unified framework, while another reviewer noted that similar ideas are standard in instruction tuning settings, thus limiting the claimed novelty.

**Reviewer Concerns:**

- **Novelty over listwise LTR**. The authors argued that the STS community largely considers a different family of losses; and that compared to ListNet, a key difference is the consideration of ranks rather than raw scores. The response also clarified that the paper's contributions include unified loading and differentiated processing as equally important, and are not limited to the loss function alone.
  - *Partially addressed*. I do agree that the authors have not just taken an existing loss and applied it to a new idea. However, I tend to think the reviewers' point about the proposal not being fundamentally novel is valid.
- **Reproducibility and sensitivity**. The authors pointed to Appendix A.8, which contains a detailed list of hyper-parameters. They also pointed to Appendix A.10, which reports sensitivity to various hyper-parameters, showing a high degree of robustness.
  - *Mostly addressed*.
- **Scope of joint optimization**. The authors pointed to results on the full CMTEB benchmark, containing a larger suite of datasets. The authors also argued that their method does not inherently favor the STS task; rather, they argued that the nature of the underlying objective for each task necessitates a different number of loss terms, and also determines the performance range for each.
  - *Partially addressed*. The CMTEB benchmark further strengthen the paper's empirical results. The discussion around the STS versus IR bias is interesting, but this issue needs more careful treatment in the paper. It is not entirely clear why the nDCG admits a much tighter loss formulation than Spearman; nor why the proposed losses help tighten the modeling of the latter. If these are standard observations in the literature, they ought to be cited.
- **Scope of baselines**. The authors pointed to results on the full CMTEB benchmark, containing more modern baselines, over which superior results are shown.
  - *Mostly addressed*.
- **Scope of performance gains**. The authors argued that the gains on the overall IR+STS score are significant, while the gap to the IR only and STS only methods are expected to be modest.
  - *Partially addressed*. The response did not seem to directly assuage the reviewer's concern that the inherent noise in evaluation is exceeded by the gains.
- **Computational cost**. The authors pointed to Appendix A.9, which showed the method does not introduce significant computational or memory overhead.
  - *Mostly addressed*.
- **Intuition for synergistic gains**. The authors provided an intuition behind the collaborative nature of the tasks, resting on the hypothesis that the STS task allows for finer-grained separation within query and document spaces.
  - *Partially addressed*. The intuition provided makes sense, though it does not appear to be woven into the narrative in the body. Since the method name emphasizes the collaborative nature of the two tasks, some discussion of why such collaboration is to be expected is important.
- **Unified framework versus task switcher**. The authors argued that the method is distinct from lossy approaches such as INSTRUCTOR, with the ability to do intelligent routing being a key part of the proposal.
  - *Mostly addressed*. I see the discussion about the "unified" terminology as not being critical; more important is whether the framework is indeed novel or useful. For this, I tend to think the author's response does make a reasonable case.

Further to the above, we offer a few comments.
- The unified data format contribution seems slightly overstated. Is this not essentially a union of the two existing formats, given that _"For fields absent in the original dataset, CoDiEmb fills them with default placeholders that are ignored during the forward pass"_.
- Section 2.2 title should perhaps be _Differentiable_ loss functions.
- Section 2.2 is overly long as is, with too many distinct ideas. Consider breaking into sub-sections.
- The PRO loss appears highly similar to the classic ListMLE loss, with a particular temperature scaling.
- It is mentioned in passing that the PRO loss is based on a "reinforcement learning loss", but no further details are provided. Does one need to perform policy gradient optimization to minimize this term? How did the authors handle this in conjunction with conventional losses?
- It is not apparent upon reading Section 2.2 why one needs all three losses, and what relative weighting is expected for them.

**Reviewer Scores:**

- **iJib**: The reviewer's comments were mostly addressed. However, given the review was not strongly in support of the paper, we think it likely the reviewer would have maintained their score at 6.
- **tR9b**: We think it possible the reviewer would have increased their score to a 4. It is possible that the reviewer could have increased it further, but given the number of detailed critiques, we tend to see this as not being overly likely.
- **CGg8**: We think it likely the reviewer would have increased their score to a 6.
- **Wzmt**: We think it likely the reviewer would have increased their score to a 4. It is possible that the reviewer could have increased it further, but given the number of critiques, we tend to see this as not being overly likely.

---

### Decision · Program_Chairs · 2026-01-26

Reject